# Ignoring correlated activity causes a failure of retinal population codes

Kiersten Ruda [1], Joel Zylberberg[2] & Greg D. Field [1✉]

From starlight to sunlight, adaptation alters retinal output, changing both the signal and noise among populations of retinal ganglion cells (RGCs). Here we determine how these light level-dependent changes impact decoding of retinal output, testing the importance of accounting for RGC noise correlations to optimally read out retinal activity. We find that at moonlight conditions, correlated noise is greater and assuming independent noise severely diminishes decoding performance. In fact, assuming independence among a local population of RGCs produces worse decoding than using a single RGC, demonstrating a failure of population codes when correlated noise is substantial and ignored. We generalize these results with a simple model to determine what conditions dictate this failure of population processing. This work elucidates the circumstances in which accounting for noise correlations is necessary to take advantage of population-level codes and shows that sensory adaptation can strongly impact decoding requirements on downstream brain areas.

[1] Department of Neurobiology, Duke University School of Medicine, Durham, NC, USA. [2] Department of Physics and Center for Vision Research, York University, Toronto, Ontario, Canada. ✉email: field@neuro.duke.edu

Population activity is the currency of sensory systems because individual neurons have limited signal capacity and variable responses to repeated presentations of the same stimuli. This variability is often shared across neurons (termed noise correlations), adding a rich complexity to the issue of information processing in neural populations[1]. There is a large body of work showing that these noise correlations can enhance or degrade signaling of sensory information, depending on the structure of noise correlations and their relationship to stimulus-evoked signals[2–7]. A crucial question is how downstream regions can best integrate signals given the noise correlations among their inputs. Perhaps ignoring noise correlations has no adverse effect on computations. On the other hand, downstream regions may need to take correlated noise into account to appropriately process their inputs. Answering this question is critical for understanding how the activity of sensory populations represents stimuli as well as generating informed hypotheses about how downstream circuits process these signals.

In the visual system, populations of retinal ganglion cells (RGCs)—the brain's sole source of visual information—exhibit noise correlations. Previous work has shown that failing to account for these correlations decreases decoded information by 0–20%[8–10]. However, these studies were performed under daylight conditions, just part of the retina's broad operating range that spans 10–12 log units of light intensity. Importantly, the structure of correlated noise changes over light intensities: correlated activity is generally stronger at lower light levels, exhibiting higher peak correlations that extend over longer spatial and temporal scales[11,12]. This shift in correlated noise across populations of RGCs raises the intriguing possibility that light adaptation changes the impact of these correlations on decoding retinal output.

To determine the impact of light adaptation and associated changes in correlated activity, we recorded from populations of rat RGCs with a large-scale multielectrode array (MEA) over conditions spanning rod-mediated (scotopic) to cone-mediated (photopic) light levels. Using a generalized linear model (GLM) to decode retinal activity, we show that at photopic light levels, accounting for correlations among RGCs improves decoding by ~20% compared to assuming independent noise among RGCs, similar to previous results in other mammals[8–10]. However, under scotopic conditions, accounting for correlations showed a significantly larger impact on decoding performance with a ~100% improvement in decoded information. Strikingly, assuming independent noise across a local population of RGCs produced poorer decoding performance than decoding with a single RGC. In this way, we demonstrate a failure in decoding neural populations when noise correlations are substantial and ignored. Importantly, these results depended on the RGC type that was analyzed, with decoding from OFF-brisk transient RGCs exhibiting greater sensitivity to correlations than decoding from OFF-brisk sustained RGCs. To generalize these results, we created a model of tuned, correlated neurons to identify conditions under which assuming independence causes decoding from the population to perform worse than decoding from a single cell. This model elucidates the circumstances where accounting for correlations not only improves visual processing, but is necessary to take advantage of population codes. More generally, this work demonstrates the large impact of context-dependent correlations in sensory processing and raises important questions about how downstream brain areas process retinal signals across light levels.

## Results

**Noise correlations are stronger at scotopic light levels.** To examine the consequences of pairwise noise correlations on retinal population codes, we recorded RGC responses across a range of light intensities from segments of rat retina on a large-scale MEA[13,14]. The retina was stimulated with spatiotemporal checkerboard noise to estimate the receptive fields (RFs), contrast response functions, and autocorrelation functions of RGCs over the MEA. RGCs were functionally classified according to their light response properties and spiking dynamics[14,15]. The results of the classification were validated by observing that each functionally defined RGC type exhibited a mosaic-like organization of RFs that approximately tiled space (Fig. 1a)[14,16,17]. We initially focus our analysis of correlated activity onto a single-cell type: OFF-brisk transient (-bt) RGCs (Fig. 1a). These cells are likely homologous to OFF parasol cells and other transient alpha-like RGCs in other mammals: they exhibit center-surround RFs, short-latency, transient light responses, and high-contrast sensitivity[14,18–20]. Focusing first on this RGC type facilitated comparing our results to previous work in the primate and rodent retina[8–10].

Understanding the role of light adaptation in retinal coding required tracking the same population of RGCs across rod-mediated (scotopic) and cone-mediated (photopic) conditions. This tracking was achieved by utilizing the electrical image (EI) of each RGC. The EI is computed from the spike-triggered electrical activity of an identified RGC across the MEA[21]. EIs serve as electrical footprints of each cell and are stable, despite changes in responses across light levels[22] (Fig. 1b). This tracking procedure was further validated by observing a nearly identical mosaic-like organization of RFs across the scotopic ($1.0\,\text{Rh}^*\,\text{rod}^{-1}\,\text{s}^{-1}$) and photopic ($10,000\,\text{Rh}^*\,\text{rod}^{-1}\,\text{s}^{-1}$) light levels examined in these experiments (see "Methods").

The pairwise noise correlations among OFF-bt RGCs were greater under the scotopic condition (Fig. 1e, f). We computed all cross-correlograms between OFF-bt RGCs responding to the white noise stimulus and estimated noise correlations by subtracting stimulus-induced correlations (see "Methods"). The area under the peak and width of the noise correlations between primary neighbors increased at the lower light level (Fig. 1e and Table 1). The spatial scale of correlations over the population of OFF-bt RGCs was also larger in the scotopic condition (Fig. 1f and Table 1). To verify that these noise correlations are not critically influenced by the white noise stimulus, we also examined noise correlations during spontaneous activity and in response to a natural movie (Supplementary Fig. 1). Consistent with previous studies[12], those measurements revealed similar magnitude noise correlations across stimulus conditions, as well as similar changes in correlation structure across light levels. Thus, the structure and magnitude of the noise correlations depended weakly on the choice of stimulus, but depended strongly on the light level at which the stimulus was presented for OFF-bt RGCs. Cumulatively, these observations indicate higher magnitude correlations that have broader temporal and spatial scales across the population of OFF-bt RGCs at the scotopic light level, in agreement with the previous work[11,12,23]. In the subsequent sections, we utilize a model-based decoding approach to determine the impact these changes in noise correlation structure have on decoding visual stimuli from populations of OFF-bt RGCs.

**RGC responses are fit well by the GLM across light levels.** The model-based decoding approach we used involves first fitting an encoding model to capture the relationship between visual stimuli and RGC spiking. This model will be inverted to estimate stimuli given RGC spike trains. Importantly, we are not claiming that this exact model inversion procedure is used in brain areas downstream of the RGCs. Since the exact computations downstream of

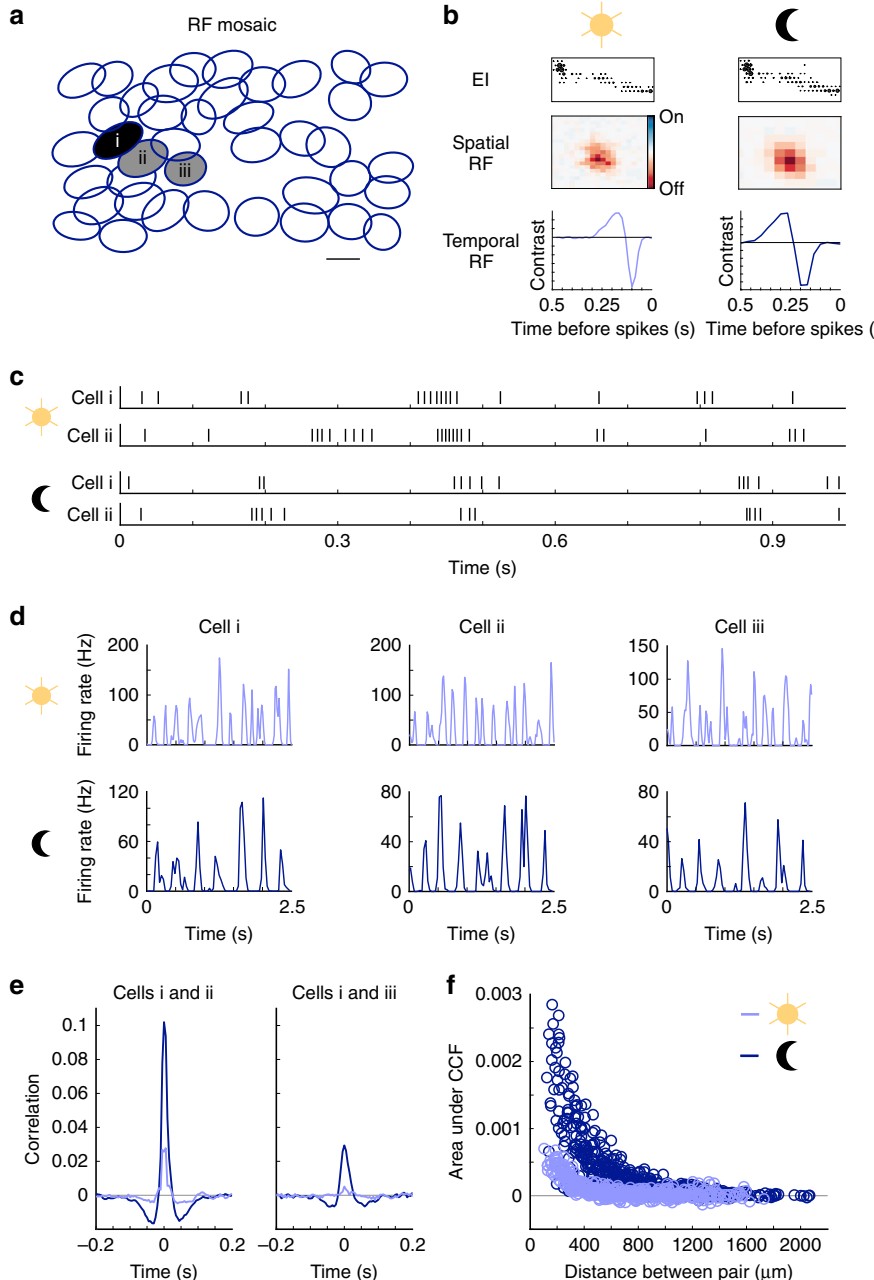

**Fig. 1 Noise correlation structure and receptive fields of retinal ganglion cells (RGCs) depend on the light level. a** Receptive field (RF) mosaic of OFF-brisk-transient (OFF-bt) RGCs. Each ellipse is the 1 s.d. Gaussian fit to an RGC's spatial RF. Scale bar is 200 μm. **b** Top row: electrical image (EI) of an example cell at two light levels, which enables tracking RGCs across light conditions. Middle row: spatial RFs at the two light levels. Bottom row: temporal RF. Both spatial and temporal integration increase in the scotopic condition. **c** Spike raster of two neighboring RGCs at photopic (top) and scotopic (bottom) light levels responding to white noise stimuli. **d** Peri-stimulus time histograms (PSTHs) in response to the white noise stimulus at each light level for the three RGCs highlighted in **a**. **e** Example noise cross-correlation functions (CCFs) across light levels of two primary neighbor cells i and ii in **a** (left) and two secondary neighbor cells i and iii (right). **f** Strength of noise correlations over pairwise distances for the OFF-bt RGC population. Each point shows the positive area under the CCF for a given pair of cells. Light adaptation causes expanded correlated noise in time and space for the scotopic condition (496 RGC pairs from 1 retina; see Supplementary Fig. 1 for correlated spiking from spontaneous firing and a natural movie stimulus).

the retina are unknown, we chose an optimal decoding approach. This procedure yields a way to estimate how well an ideal downstream system could estimate the stimulus, given the RGC spike trains using different assumptions about correlations between cells[1,9].

To quantitatively describe RGC spiking in response to a checkerboard stimulus, we use the GLM, a phenomenological model for retinal encoding that can also be used for Bayesian decoding[9]. The GLM transforms visual stimuli to spike times by

first filtering the stimulus through the spatiotemporal RF and applying a spike history filter to account for refractoriness and spike bursts (Fig. 2a). This signal is then passed through a static nonlinearity to yield a predicted firing rate, and spike times are generated with a Poisson process. We first fit OFF-bt RGCs with an independent version of the GLM, in which each cell is fit individually and the spiking of one RGC is independent of the other RGCs (except for stimulus-induced correlations). The same cells were fit at each light condition separately to optimize model

**Table 1 Measurements of correlation structure across light levels for the OFF-bt and OFF-bs RGCs.**

|  | CCF area | CCF width | Correlation spatial scale (μm) |
|---|---|---|---|
| OFF-bt photopic | 0.0005 ± 0.00002 | 0.04 ± 0.0006 | 217 ± 6 |
| OFF-bt scotopic | 0.0012 ± 0.00005 | 0.047 ± 0.001 | 290 ± 6 |
| OFF-bt change over light levels | 0.0008 ± 0.00005 | 0.008 ± 0.001 | 76 ± 16 |
| OFF-bs photopic | 0.0001 ± 0.00001 | 0.033 ± 0.001 | 170 ± 12 |
| OFF-bs scotopic | 0.0003 ± 0.00001 | 0.035 ± 0.001 | 195 ± 8 |
| OFF-bs change over light levels | 0.0001 ± 0.00001 | 0.001 ± 0.002 | 39 ± 25 |
|  | **P values** | **P values** | **Fraction of retinas** |
| OFF-bt: photopic vs. scotopic | $P \ll 0.001$ | $P \ll 0.001$ | 3/3 |
| OFF-bs photopic vs. scotopic | $P \ll 0.001$ | $P = 0.59$ | 2/3 |
| Photopic: OFF-bt vs. OFF-bs | $P \ll 0.001$ | $P \ll 0.001$ | 3/4 |
| Scotopic: OFF-bt vs. OFF-bs | $P \ll 0.001$ | $P \ll 0.001$ | 3/3 |
| Light-level change: OFF-bt vs. OFF-bs | $P \ll 0.001$ | $P < 0.005$ | 1/3 |

Values are mean ± s.e.m. All data comes from four retinas for the photopic condition and three retinas for the scotopic condition (three retinas in common between conditions). Sample sizes, in the number of primary pairs, are OFF-bt photopic, 171, OFF-bt scotopic, 121, OFF-bt change over light levels, 121, OFF-bs photopic, 124, OFF-bs scotopic, 92, OFF-bs change over light levels, 92. Correlation spatial scales were found by fitting an exponential to a cell type and light level separately for each retina. The fraction of retinas column reports in how many retinas the spatial scales being compared did not overlap at the 95% confidence intervals of their respective fits.

performance at each light level. The independent GLM predicted held-out responses well at both light levels, as measured by the explained variance in firing rates (photopic: 0.59 ± 0.01, mean ± s. e.m., 100 cells from four retinas, scotopic: 0.58 ± 0.01, 69 cells from three retinas; Fig. 2c, d). Furthermore, the GLM captured changes known to occur in light adaptation, such as larger spatial RFs and slower temporal integration[24,25] (Fig. 1b and Supplementary Fig. 2).

To account for noise correlations between RGCs and determine their impact on decoding, we separately fit a coupled version of the GLM. The coupled GLM includes pairwise coupling filters so that the activity of one RGC can influence the responses of other RGCs, allowing the coupled GLM to capture noise correlations in RGC activity[9]. Because correlations decrease rapidly with the distance between pairs of cells (Fig. 1f), we used local groups of RGCs in the coupled GLM, choosing each group based on a central RGC and all of its recorded neighbors (Fig. 2b). For single-cell PSTHs, the coupled GLM predictions and performances were very similar to those of the independent GLM at both light levels (Supplementary Fig. 2). The coupled model predicted noise correlations well, while the independent model did not (Fig. 2e, f), consistent with the previous results[9]. This indicates that the independent GLM captures signal correlations across the population of RGCs, but the coupled GLM is necessary for capturing the signal and noise correlations. Having established the GLM as an accurate description of RGC activity under scotopic and photopic conditions, we next use the independent and coupled versions of the GLM to probe the impact of noise correlations on decoding retinal output over light levels.

**Ignoring correlated noise severely impacts scotopic decoding.**
We estimated white noise stimuli from recorded responses to elucidate the impact of noise correlations on decoding OFF-bt RGC output. To perform model-based decoding of responses, we inverted the independent and coupled GLMs fit to recorded OFF-bt RGCs (see Fig. 3). We compared the decoding performance between these two models to determine the extent to which ignoring noise correlations between RGCs diminished decoding performance. We performed Bayesian decoding, which optimally extracts stimulus information available in the RGC response structure that is captured by the GLM[9]. Given a set of spike times from a local group of RGCs, we decoded the intensity of a single stimulus pixel over six sequential frames. For this analysis, the stimulus pixels and RGCs were chosen such that the pixel was

predominantly covered by the center-most RGC of the group of cells (see "Methods"). We report decoding performance with a signal-to-noise ratio (SNR), which quantifies the mutual information rate in bits s$^{-1}$ that the decoded estimate provides about the actual stimulus[26].

At the photopic light level, the coupled GLM is a more accurate decoder, providing 22 ± 3% (mean ± s.e.m.) more information than the independent GLM over all groups of OFF-bt RGCs (Fig. 4a, b; 55 groups of RGCs from four retinas). However, at the scotopic light level, the importance of correlations for accurate decoding substantially increased for OFF-bt RGCs. Accounting for noise correlations with the coupled GLM provided 105 ± 18% more information than the independent GLM (Fig. 4b; 37 groups of RGCs from three retinas, the difference over light levels $P \ll$ 0.001). Furthermore, the improvement in decoding for a given group of cells correlated positively with the strength of noise correlations in that group, indicating that accounting for correlated activity enhances decoding most when correlated noise is largest (Fig. 4c).

**Population failure: single RGCs can outperform populations.**
To better understand the significance of the information loss due to ignoring noise correlations, we compared the decoding performance of the independent GLM to that of the best-performing single-cell model. The single-cell model was simply the individual GLM for the RGC centered over the decoded pixel. This comparison allowed us to relate the cost of ignoring noise correlations among a local population of RGCs to the benefit of decoding using responses of more than one RGC. Note that here the independent GLM eliminates noise correlations across the population of RGCs, but signal correlations are preserved. Surprisingly, in many of the tested groups, the single-cell GLM outperformed the independent population GLM (Fig. 4a). We call this effect population failure because the GLM fit to a population of RGCs decodes less information than from a single RGC when the population is assumed to be noise independent.

Population failure occurred at both light levels, but more frequently in the scotopic condition (Fig. 4d). At that light level, the majority of groups of RGCs exhibit this population failure mode (83 ± 6%, the mean frequency of population failure ± s.e.m., 37 groups of RGCs, three retinas), and among those groups the single-cell GLM provided 75 ± 23% more information than the independent GLM (mean ± s.e.m, Fig. 4e). However, in the photopic condition, about half of the groups (50 ± 6%, mean frequency of population failure ± s.e.m., 55 groups of RGCs)

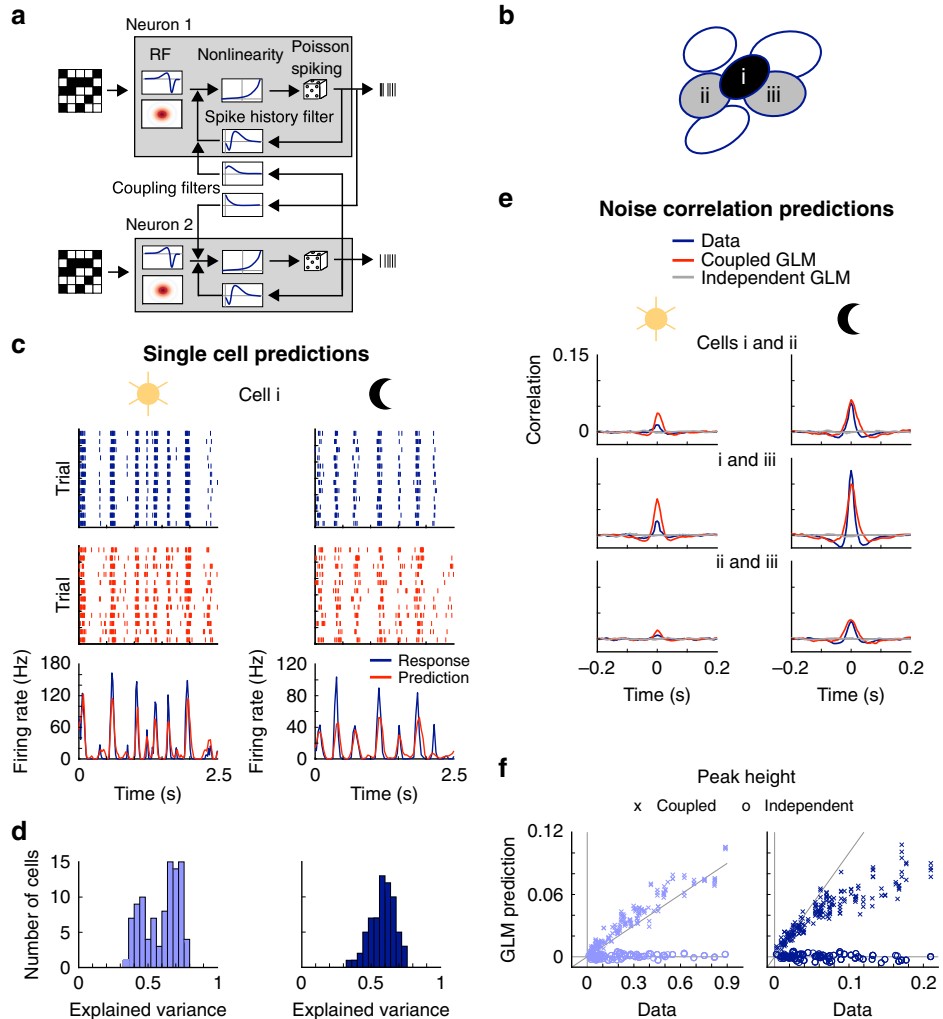

**Fig. 2 A generalized linear model (GLM) captures retinal ganglion cell (RGC) responses and pairwise correlation structure. a** GLM diagram for two coupled cells, adapted from Pillow et al.[9]. The stimulus is filtered by a receptive field (RF), this generator signal is passed through a nonlinearity to generate firing rates, and then stochastic spikes are created. Spike history and coupling filters (for the coupled GLM) also influence the generator signal. **b** Example local group of RGCs to which the GLM model is fit. **c** Example recorded raster (blue), GLM predicted raster (red), and peri-stimulus time histogram (PSTH), and predicted PSTH (bottom) for cell i in **b**. The left column shows the photopic light level and the right column shows the scotopic light level. **d** Distribution of explained variances for the GLM predicted PSTHs for photopic (left) and scotopic (right) light levels (photopic:100 OFF-bt RGCs from four retinas; scotopic: 69 RGCs from three retinas). **e** cross-correlation functions (CCFs) and GLM predicted CCFs of cell pairs shown in **b** at two light levels. **f** Peak height of predicted CCFs compared to measured CCFs. The coupled GLM predicts close to the measured noise correlation values, while the independent GLM predicts no noise correlations (left: photopic condition; right: scotopic condition; 189 pairs from 11 groups of RGCs from one retina).

showed moderate population failure ($19 \pm 4\%$ more information than the independent GLM, mean $\pm$ s.e.m). Notably, the single-cell GLM uses the exact same parameters as its corresponding cell in the independent GLM, so our findings are not a consequence of model fitting issues. Rather, this result demonstrates that decoding under the assumption that a population of RGCs is independent can be so suboptimal that it extracts less information than a single cell. This population failure under the assumption of independence is a striking example of the importance of accurately accounting for correlations in processing population activity, particularly in scotopic conditions.

We next performed a series of controls to assess how particular details of our decoding analysis might influence these results. In the analyses above, we chose local groups of cells based on a central RGC with its nearest neighbors. Thus, the RFs over the population of RGCs had some overlap with the decoded stimulus pixel so that each cell provided nonzero decoding information about that pixel intensity (e.g., Figure 4a; note RF outlines are

plotted at a 1 s.d. contour of a Gaussian fit, so the RFs extend well beyond the RF outline). To determine how this choice of population impacts decoding, we also decoded using larger groups of cell clusters, including secondary and tertiary neighbors. Including RGCs with RFs far away from the decoded stimulus pixel did not significantly alter the performances of the coupled or independent GLMs because those cells contribute minimal information to decoding and do not exhibit strong correlations with RGCs close to the decoded stimulus pixel, as expected (Supplementary Fig. 3). Thus, our selection of local groups of RGCs is not a crucial factor in the role of correlations for decoding.

We further sought to ascertain whether population failure generalizes beyond temporal decoding by instead decoding spatial patterns of stimulus pixels for one movie frame. Under this decoding task, the coupled GLM continues to perform substantially better than the independent GLM at the scotopic light level ($52 \pm 16\%$, mean $\pm$ s.d. over bootstraps; Supplementary Fig. 4). In

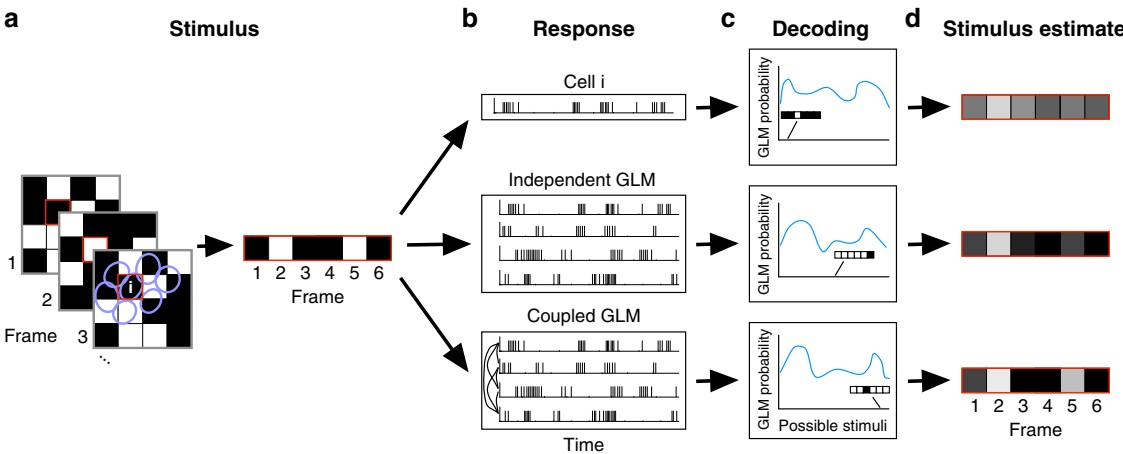

**Fig. 3 Schematic showing generalized linear model (GLM)-based Bayesian decoding. a** One stimulus pixel, highlighted in red, is chosen over several frames (left), yielding a sequence of intensity values (right). **b** The corresponding response of a single retinal ganglion cell (RGC) (top) or population of RGCs (middle and bottom) are extracted. **c** The probability of each possible stimulus given the input response is computed under a GLM fit to that population. **d** Summing over the possible stimuli weighted by their probabilities gives the optimal Bayesian estimate of the stimulus. In general, each GLM provides a different estimate of the original stimulus.

addition, the independent GLM decodes less information than smaller groups of coupled RGCs, exhibiting a form of population failure because 18 cells in the independent GLM perform worse than 7 cells in a coupled GLM (Supplementary Fig. 4G, H). These results demonstrate that the cost of ignoring correlations is a general feature of spatial and temporal decoding from OFF-bt RGCs.

Finally, to verify that changes in correlation structure causally affect the difference in decoding performance between coupled and independent GLMs, we simulated RGC-population responses with the GLM and then used the GLM to decode these simulated responses. As we observed when decoding measured responses at the scotopic light level, we hypothesized that stronger coupling among neurons would lead to a higher percent improvement in decoding SNR when accounting for noise correlations versus assuming independence. Indeed, a larger correlation strength between RGCs causes the coupled decoder to perform much better than the independent decoder (Supplementary Fig. 5). This simulation emphasizes how the amount of correlated noise impacts decoding performance under the independence assumption.

**The cost of ignoring correlations depends on RGC type**. We next investigated the extent to which the population failure phenomenon occurs in a distinct RGC type, the OFF-brisk sustained (-bs) RGCs. These cells likely correspond to RGCs called OFF delta or OFF sustained alpha cells in other studies[14,19,20]. The correlation structure across the OFF-bs RGC population shows that the magnitude, timescale, and spatial scale of correlations is smaller than in OFF-bt RGCs (Fig. 5a and Table 1). In addition, the correlations among OFF-bs RGCs do not change with light adaptation as much as in OFF-bt RGCs (Table 1). To determine the role of accounting for correlations in decoding OFF-bs RGC activity, we next compared independent and coupled GLM decoders fit to groups of OFF-bs RGCs (Fig. 5). Accounting for correlations only improved decoded SNR by 2.9 ± 0.7% (mean ± s.e.m.) in the photopic condition and 4.4 ± 0.8% in the scotopic condition (Fig. 5c; photopic: 37 groups of RGCs from four retinas, scotopic: 20 groups of RGCS from three retinas, the difference over light levels $P = 0.19$). While some single-cell GLMs decode better than the independent GLM, the frequency and amount of this population failure under the independence

assumption were much smaller than in OFF-bt RGCs (Fig. 5e, f; photopic: frequency of population failure = 19 ± 5%, mean ± s.e.m., % improvement when there is population failure = 1.9 ± 1%, scotopic: frequency of population failure = 28 ± 8%, % improvement when there is population failure = 0.7 ± 0.2%). These results demonstrate that the role of noise correlations in decoding RGC activity depends on both adaptation state and cell type.

**A simple geometric model reproduces population failure**. Assuming noise independence among OFF-bt RGCs frequently caused population failure, particularly under the scotopic condition (Fig. 4d, right). To better understand this potentially counterintuitive result, we utilized a previously developed geometric visualization of noise correlations and decoding performance[1] (Fig. 6). The goal of this analysis is to determine if population failure is possible in a simple, toy model, and to develop some intuition for when and why it happens. We analyze a larger and more realistic model of our retinal population below (see Fig. 7). For this first analysis, we created a simplified model of two neurons responding to two different stimuli. The neurons respond to those stimuli with different mean firing rates, and their responses on any given trial are given by those mean responses plus Gaussian noise with variance equal to mean (i.e., Poisson variability). This noise was correlated between the two cells, and we varied the degree of correlation in our analysis. Given these neuronal responses, we used the d-prime[2] metric to quantify how well the two neurons' responses could discriminate between the two stimuli[27].

We first considered the case where both signal and noise correlations are positive, which is the most common occurrence when considering nearby RGCs of the same type. In Fig. 6a, the two model cells exhibit strong noise correlations, causing their joint response distributions to have an elliptical shape (Fig. 6a left, green and blue solid ellipses). The optimal decoder (red line) accurately discriminates the two stimuli (Fig. 6a right, red bar). However, if the noise is assumed to be independent between the two cells (Fig. 6a left, green and blue dashed circles), the resulting decoder is nearly orthogonal to the optimal decoder (compare black and red lines). In this case, the independent noise assumption causes a large decrease in decoding performance (Fig. 6a right, gray bar). For comparison, we discriminated the two stimuli using just the responses of cell 1 (shown in Fig. 6e). In this example with strong

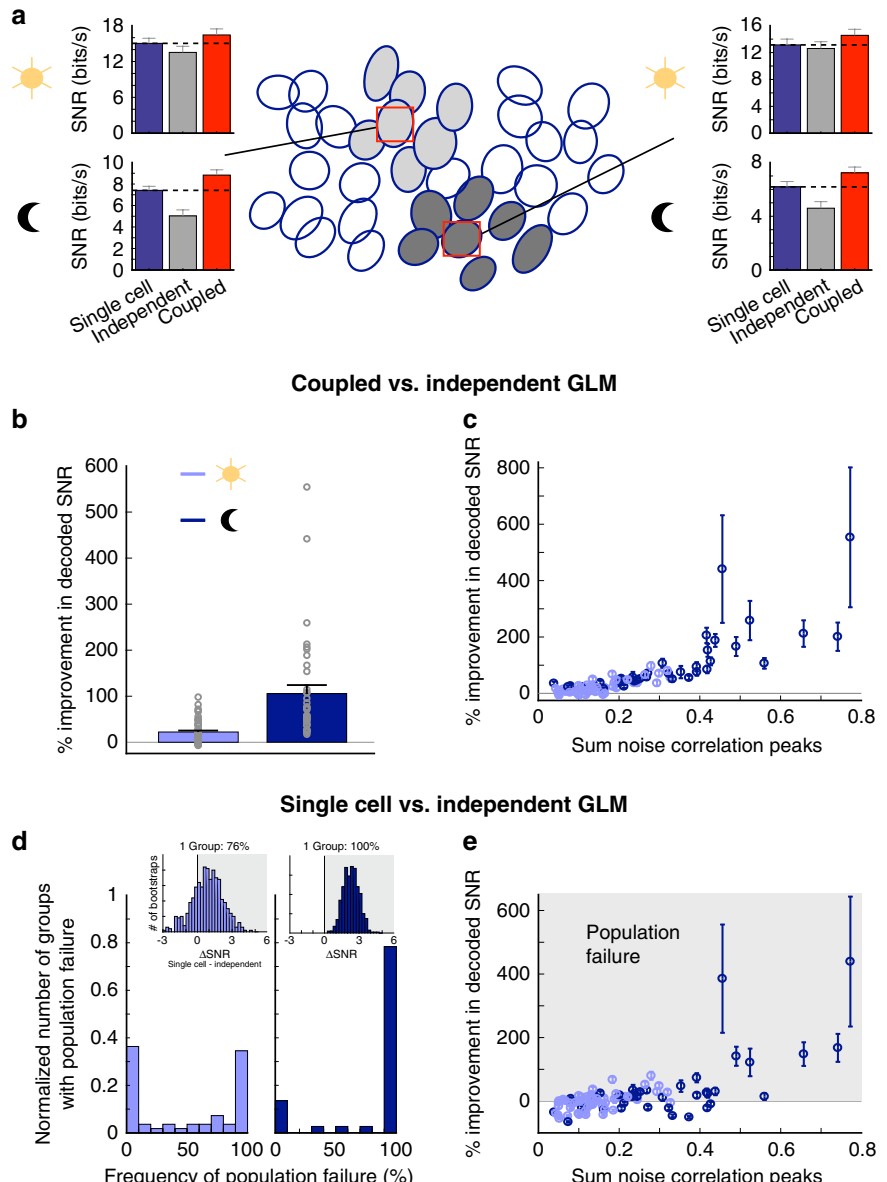

**Fig. 4 Assuming noise independence yields poor decoding performance under scotopic light levels and can perform worse than decoding individual retinal ganglion cells (RGC) responses. a** Decoding examples for two groups of OFF-bt RGCs. In these examples, the coupled and single-cell GLMs decode better than the independent GLM at scotopic light levels: see dashed lines in bar plots. Error bars (s.d.) are from bootstrapping decoded signal-to-noise ratio (SNR). **b** Average percent improvement in decoded SNR at each light level when using the coupled GLM relative to the independent GLM. Error bars are s.e.m. **c** Percent improvement in decoding relates positively to the amount of noise correlation in the groups of RGCs ($R^2 = 0.6$ for a single-term exponential). Noise correlation for each group of RGCs is quantified by accumulating the peaks of cross-correlation functions (CCFs) between the centered RGC and its neighboring cells. Error bars are s.d. from bootstrapping. **d** Insets: for one of the groups of RGCs in **a**, the difference in decoded SNR between the single-cell and independent GLMs was computed over 500 bootstraps. The distribution of these SNR differences is plotted at each light level (left and right). The percentage of samples where this SNR difference is positive yields the population failure frequency for each group of cells. Main panel: distribution of population failure frequencies over all groups of RGCs at the cone (left) and rod (right) light levels. **e** Population failure relates positively to the number of pairwise noise correlations across RGCs in each decoded group. Error bars are s.d. from bootstrapping. For panels **b**–**e**: photopic: 55 groups of RGCs from four retinas, scotopic: 37 groups of RGCs from three retinas (three retinas in common between conditions).

noise correlations, the single cell outperforms the two-cell decoder that assumes independence among the cells (Fig. 6a right, blue bar). Thus, the phenomenon of population failure can be recapitulated in a simple example consisting of two cells and linear discrimination of two stimuli.

In Fig. 6b, we examine this two-cell model with weaker noise correlations. In that case, the decoder under the independence assumption was very similar to the optimal decoder (Fig. 6b, left, black and red lines). Consequently, assuming independence

causes a very small decrease in decoding performance, and the independent decoder outperforms the single-cell decoder (Fig. 6b, right, gray and blue bars).

Next, we repeated our analysis over a wide range of possible signal and noise correlation values, revealing a number of states in this simple model that exhibit population failure (Fig. 6c, d, f–h). Specifically, the system produces population failure when signal and noise correlations have the same sign (termed the sign rule, popularized by Averbeck et al.[1]), and the noise correlations

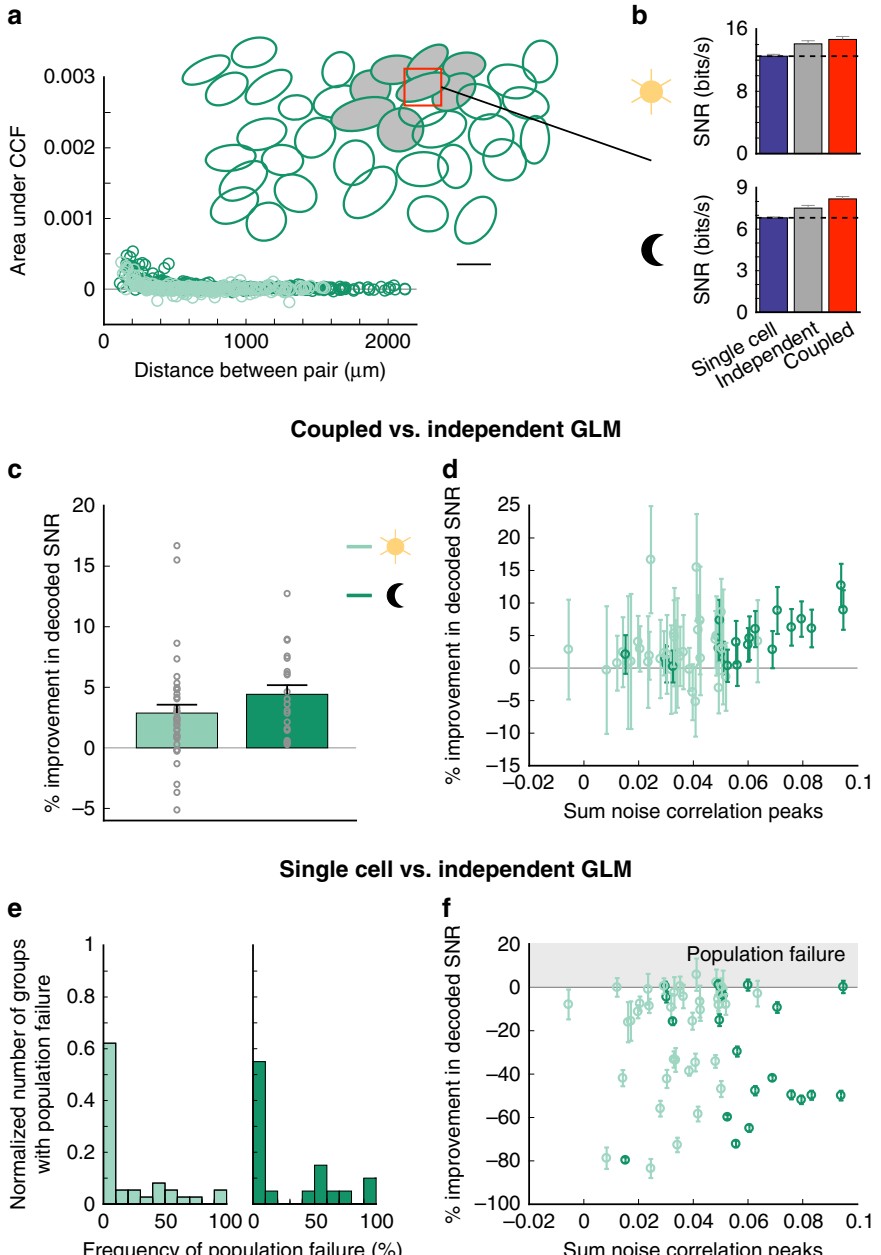

**Fig. 5 The cost of ignoring correlations depends on retinal ganglion cell (RGC) type. a** Correlation structure of OFF-bs RGC population across light levels (left) and RF mosaic (right) in the same retina as Fig. 1 (351 RGC pairs). Scale bar is 200 μm. **b** Decoding result for an example group of cells over light levels. In this instance, the independent and coupled GLMs perform similarly. Error bars are s.d. from bootstrapping. **c** Average percent improvement in decoded SNR across light levels when using the coupled GLM over the independent GLM. Note the compressed y axis compared to Fig. 4b. Error bars are s.e.m. **d** Relationship between percent improvement in decoding and summed noise correlations between a centered RGC and its neighboring cells. Note the compressed x axis compare to Fig. 4c. Error bars are s.d. from bootstrapping. **e** Distribution of population failure frequencies. **f** Population failure as a function of the amount of noise correlation in a group of RGCs. Error bars are s.d. from bootstrapping. For panels **c–f**: photopic: 37 groups of RGCs from four retinas, scotopic: 20 groups of RGCs from three retinas (three retinas in common between conditions).

are strong (Fig. 6g, purple zones). While we have only considered correlations among two OFF types in our retinal recordings, it is important to note that negative signal correlations and negative noise correlations are expected when decoding responses from ON and OFF RGCs with overlapping RFs[11,12]. Thus, real RGC pairs tend to have noise and signal correlations of the same sign, predicting population failure when noise correlations are strong.

In summary, this model demonstrates how population failure can occur in a very simple system—where decoding from a single neuron performs better than decoding from two neurons that both convey nonzero stimulus information. We emphasize that

we view this simple model as a tool for helping to build intuition and not as a quantitative replication of our experimental results. In the next section, we examine decoding performance in a population model that more closely resembles the conditions in our experiments.

**Signal and noise conditions dictating population failure.** To investigate the conditions under which a single RGC can outperform a population that is assumed to be independent, we modeled our experimental findings by simulating the responses of

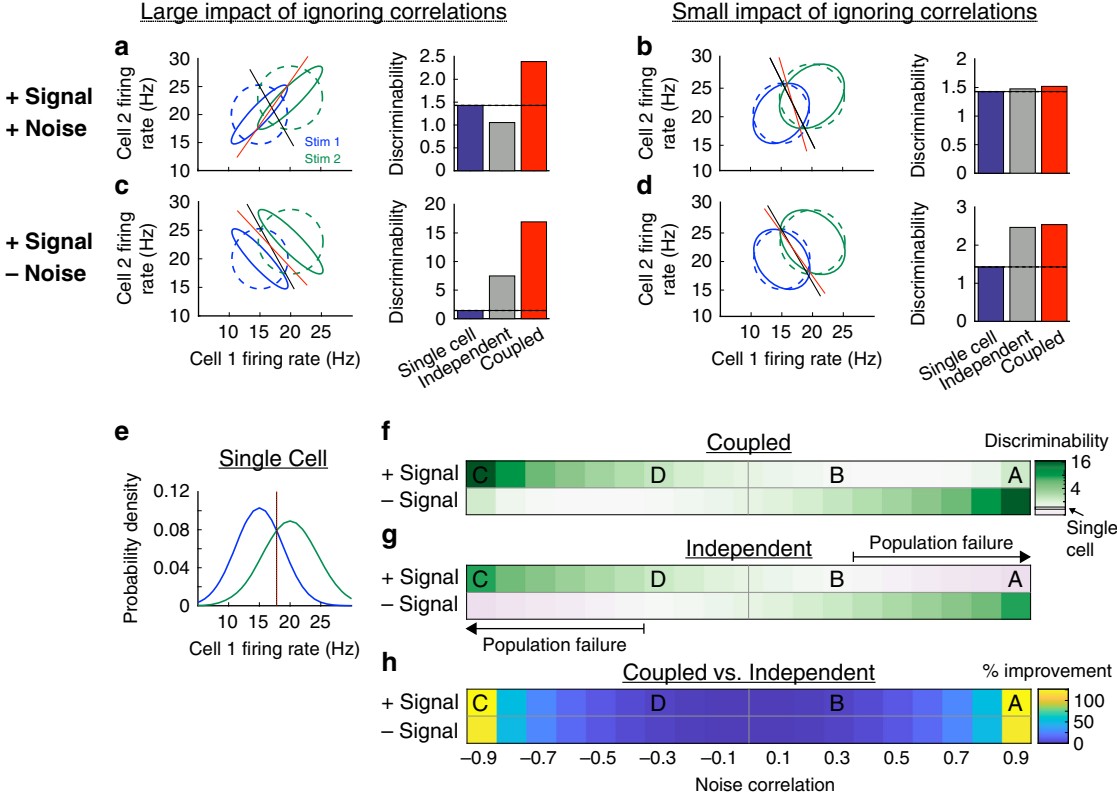

**Fig. 6 A simple, two-cell model can reproduce population failure. a** Left: joint responses when correlated activity is taken into account (solid ellipses) and when it is ignored (dashed ellipses). Geometrical perspective on decoding adapted from Averbeck et al.[1]. The optimal decoder with knowledge of correlations (red line) is very different than the decoder without knowledge of correlations (black line), resulting in high error for the independent decoder. Right: the independent decoder (left, black line) discriminates the two stimuli worse than the coupled decoder (left, red line) and the single-cell decoder (cell 1, **e**). **b** Same as **a** but with low-noise correlations. **c**, **d** Same as **a**, **b**, respectively, with opposite signs between signal and noise correlations. Here, the independent decoder is more similar to the optimal decoder, resulting in similar discrimination performance. **e** Response distributions of cell 1 in panels **a**–**d**, showing that the optimal decoding line for a single-cell model lies at $x = 18$ Hz. **f** Heatmap of discriminability for the coupled decoder over all possible signs of signal and noise correlations, as well as noise correlation magnitudes. **g** Same as **f**, but for the independent decoder. Purple regions indicate states in which the two-cell decoder assuming independence performs worse than decoding from cell 1. **h** Heatmap of a percent improvement in discriminability between the coupled and independent decoders.

a two-dimensional grid of RGCs (Fig. 7a). Each RGC response was given by applying its RF to the stimulus as a linear filter, plus additive noise. This noise was Gaussian-distributed with variance equal to the mean (i.e., Poisson noise), and we varied the degree to which this noise was correlated between neurons. To match our experiments, we simulated the responses of these neurons to binary white noise stimuli. We arranged the RFs in a hexagonal grid to approximate the mosaic of one RGC type, with the relationship between RF and stimulus pixel sizes set similarly to those in our experiments. Within this model, we used discriminability (d-prime[2]) to quantify how well intensity values in the central stimulus pixel could be discriminated given the neural responses[27]. As with our GLM-based approach, we compared the performance of a decoder that accounts for correlations among cells (coupled decoder), one that assumes independent noise among cells (independent decoder), and one that just uses the responses of one RGC (single-cell decoder). We systematically varied three parameters that determine signal and noise correlation structure across this population: peak noise correlation strength, the spatial scale of noise correlations, and RF overlap (Fig. 7a). This process enabled us to map out the conditions under which population failure arises for neural populations similar to those in our experiments.

We found that, when there is little overlap between RFs and noise correlations are present, the independent decoder often

discriminates the stimulus worse than the single-cell model (Fig. 7d, top several rows, purple areas). The cost of assuming independence in the population becomes more severe as the noise correlations are made stronger and/or broader. As RF overlap increases, the independent decoder performs better than the single-cell decoder when noise correlations have a small spatial scale and magnitude (Fig. 7d, left columns). This improved performance results from neighboring cells providing more signals about the intensity of the decoded pixel (signal correlations across the population increase), which overcomes the errors due to ignoring small noise correlations. However, ignoring larger and broader noise correlations eventually outweighs this advantage, resulting in more extreme population failure (up to 50% less discriminability than the single-cell decoder for the parameters we explored; Fig. 7d, right columns). Note that the coupled decoder discriminates much better than the independent decoder in the presence of strong and broad noise correlations (Fig. 7e).

We also examined the performance of this model when the stimulus consisted of binarized natural scenes (Fig. 7f–k). This stimulus set contains more power at lower spatial frequencies than the checkerboard noise (Fig. 7h). The frequency and conditions under which population failure occurred in this model were nearly identical between checkerboard and natural stimuli, suggesting that our population decoding results

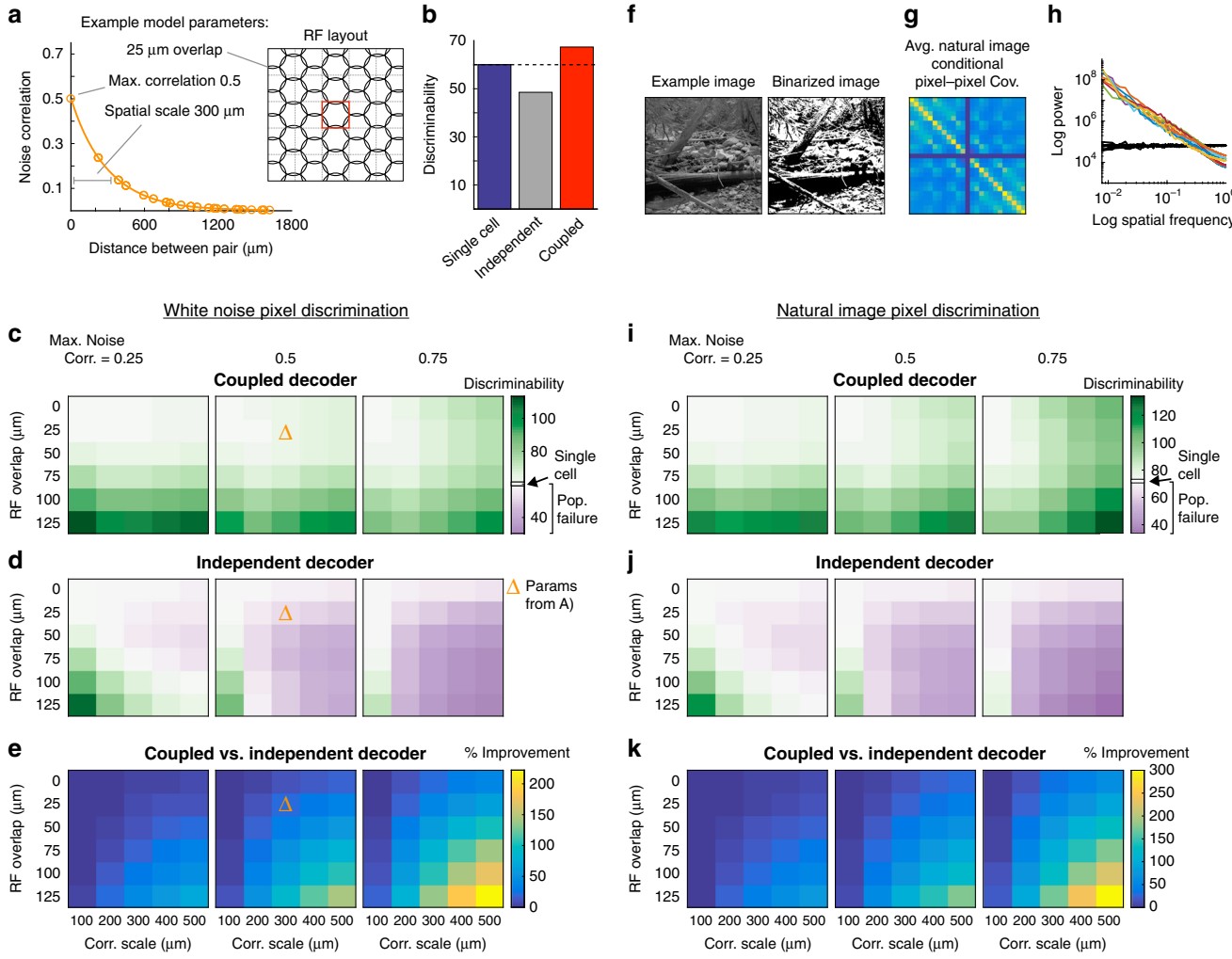

**Fig. 7 RF overlap, peak correlation, and spatial extent of correlations dictate the conditions for population failure. a** Correlation structure (left) and receptive field (RF) mosaic (right) of an example model with the given RF overlap, maximum correlation, and spatial scale of correlations. **b** Decoding results from the model parameters shown in **a** illustrate a state in which the assumption of independence causes the population to perform worse than a single cell. **c** Heatmap of discriminability for the coupled decoder over all values for RF overlap, maximum noise correlation, and spatial scale of noise correlations. **d** Same as **c**, but for the independent decoder. As the magnitude and scale of noise correlations increase, the independent decoder discriminates less information. Purple regions indicate population failure states. **e** Percent improvement in discriminability achieved by the coupled decoder over the independent decoder. **f–k** The results from the same model using natural image stimuli. **f** Example natural image that was used to estimate the stimulus statistics for the natural images. Original images (left) were binarized (right), and the means and covariances were then estimated on 5 × 5 pixel patches, conditioned on the state of the central pixel. **g** Pixel–pixel covariance of the binarized natural images. The covariance conditioned on a white center pixel was averaged with the covariance conditioned on a black center pixel (hence the zero variance in the middle pixel). **h** Spatial power spectrum of binarized natural images (colored lines) and randomly sampled white noise images (black lines). The natural images exhibit more power at low spatial frequencies, while the white noise images have equal power over spatial frequencies. **i–k** Same as **c–e** with natural image stimulation.

generalize well between these stimulus types (Fig. 7c–e compared with Fig. 7i–k).

The simplified model we present here highlights that accounting for correlated noise is most important for decoding the stimulus when correlations are large, in agreement with our experimental findings. By considering the transition into population failure modes based on noise correlation parameters, this model demonstrates how changing the correlation structure in OFF-bt RGCs across light levels can alter the frequency and magnitude of population failure. The correlations in OFF-bs RGCs, however, are generally too small at both light levels for population failure to occur.

These modeling results also reproduce a point about the limits that noise correlations can place on a neural system. Focusing

only on the coupled decoder, there are several conditions where high noise correlations limit the discriminability of the neural population compared to weak noise correlations (Fig. 7c, bottom rows). This phenomenon has been previously described, and since we are primarily focused on the consequences of assuming independence given the presence of noise correlations, we do not consider it further[3].

## Discussion

A major question in early vision is how circuits downstream of the retina process the visual information conveyed by populations of RGCs. Central to this question is the impact of correlated activity among RGCs, which can be a significant factor in neural computations depending on the context. Here, we examine how

light adaptation alters the role of correlations in decoding visual stimuli from RGC populations in the rat retina. We find that under moonlight conditions, decoders assuming independent noise among OFF-bt RGCs recover much less visual information than decoders that account for pairwise noise correlations. This reduction in performance can be so large that decoders assuming independence perform worse than decoding from a single RGC (Figs. 4, 6, and 7). We call this state population failure because decoding the population fails to reach the performance of a single cell. Accounting for correlations, however, avoids this state and enables decoders to benefit from population codes. We use a simple model to demonstrate how the structure of activity correlations determines the cost of assuming responses with independent noise, accounting for why our results depend both on light level and RGC type. These findings raise several questions about the role of correlations in adaptation and visual processing that we discuss below.

**Comparison to previous studies, interpretation, and caveats.** The importance of correlated activity is a much-debated topic in vision research[4,8–10,28–34]. Previous studies have examined the role of correlated spiking in both visual encoding and decoding, yielding a range of conclusions. For decoding, ignoring signal correlations in large populations of RGCs is particularly detrimental to decoding performance, especially when decoders have access to the fine temporal structure in spike trains[34]. Here, however, we focus on the role of noise correlations. Other studies examining correlated noise have concluded that between 0 and 40%, more information is available when decoders account for noise correlations[4,8–10]. Our results are most comparable to the Pillow et al.[9] and Meytlis et al.[10] studies because they analyzed similar population sizes and employed the same GLM-based decoding strategy, although the cell types and species used in all three studies were distinct. The decoding improvement we find at the photopic light level agrees relatively well with their 20% and 13% results, respectively. Our study departs from this previous work by determining how this decoding improvement depends on adaptation state and cell types that encode distinct visual features (OFF-bt vs. OFF-bs RGCs). The effect of light level on OFF-bt RGCs is particularly striking: decoded information can be doubled by accounting for correlations. This improvement is a substantially larger effect than previous results at photopic light levels, illustrating the potent impact of light adaptation on retinal output.

We focused on two OFF RGC types because they exhibited well-isolated spikes and relatively complete RF mosaics, two features necessary for this analysis. Functionally similar ON RGCs (ON-bt and ON-bs) were also present in our data: These ON types exhibited correlated activity magnitudes that were in between the two OFF types, and diminished decoding performance when correlations were ignored for the ON types were similarly bounded by the results of the two OFF types.

How could accounting for correlations improve retinal decoding? One possibility is that correlated activity conveys visual features that are unavailable from individual responses, such as fine spatial features at the intersection between two RFs[4,35,36]. To check for this possibility, we analyzed synchronous spike-triggered averages from pairs of RGCs. We did not find evidence that synchronous spikes provide a higher acuity representation of visual space (Supplementary Fig. 6). An alternative possibility is that accurate decoding requires an accurate model of the noise in RGC populations[1]. When correlated noise is large and spatially extensive, such as for OFF-bt RGCs at scotopic light levels, assuming independence is the wrong noise model, and this assumption diminishes decoding

so much that performance can fall below that of decoding from a single cell.

A simple intuition for the population failure effect can be achieved by considering the following situation. If a single OFF-bt RGC generates a brief volley of spikes, a decoder will interpret this response as resulting from a transient decrease in light intensity. If all the OFF-bt RGCs around that cell also generated spikes, the decoder will estimate a large decrease in light intensity because many cells were driven to spike together. However, this interpretation may only be correct if the OFF-bt RGCs are acting independently. If the decoder knows the cells exhibit strong noise correlations, then it should discount this conclusion in favor of a smaller decrease in light intensity.

Many of the findings presented here are based on GLM fits to the responses of RGC populations, raising the possibility that at least some of these conclusions are model-dependent. The GLM captures a majority of the response variance to white noise stimuli (up to 80%), but remains an imperfect model of RGC encoding[37,38]. This mismatch between model and data is likely to impact the quantitative estimates that we and others have made on the cost of assuming RGCs are independent[9,10]. Of particular concern is whether assuming independence in a population of RGCs can actually yield worse performance than decoding a single RGC. To address this issue, we also utilized a more general examination of how correlations can impact decoding. Figure 6 demonstrates that population failure is certainly possible, even with simple linear decoders. Figure 7 shows that this effect depends on the amount of RF overlap and strength of noise correlations, both of which change with light level. This simplified model has many differences from our data, with uniform, circular RFs, uniform RF overlap, firing rates that depend linearly on the stimulus, and decoding using discriminability (d-prime$^2$) rather than GLM-based Bayes-optimal stimulus estimation. Nevertheless, the simplified model in Fig. 7 reproduced the trends in our data. Furthermore, we show that population failure can occur when decoding spatial stimulus patterns from RGC responses (Supplementary Fig. 4), indicating that these results are not specific to temporal decoding. Together, these analyses show that ignoring strong correlations can degrade decoding and reproduce population failure in a manner that does not depend strongly on the details of the decoding task or the precise nature of the RGC output.

**Generalization to more natural stimulus and decoding conditions.** We have primarily focused on decoding the temporal sequence of one pixel in a checkerboard noise stimulus, raising the question of the extent to which these results generalize to more natural stimulus and decoding conditions. Given our findings that the strength of noise correlations dictates the impact on different types of decoders (Figs. 6, 7, and Supplementary Fig. 5), one important check is that the magnitude of the noise correlations remains similar between white noise and natural movies. We compared noise correlations between checkerboard white noise and a naturalistic movie (camera mounted on a cat's head roaming through the forest; Supplementary Fig. 1); the correlations depended weakly on the stimulus and when they differed, noise correlations were higher for the natural movie. In combination with our modeling findings in Fig. 7f–k, this result suggests that accounting for noise correlations is no less important when decoding RGC responses to natural stimuli.

Note that one previous study examined the decoding of natural scenes and concluded that ignoring noise correlations had minimal impact on decoding performance[10]. Importantly, the RGC types analyzed in that study were not characterized or clearly defined. We show here that the importance of noise

correlations for decoding RGC responses depends strongly on the cell type that is used in the decoding task, as well as the adaptation state of the retina.

Finally, we have benchmarked the cost of ignoring noise correlations among a population of neurons against the performance of decoding the responses of a single neuron. However, the phenomenon of population failure can be framed somewhat more generally: We illustrate one such case in Supplementary Fig. 4F–H, where the task is to decode a spatial (rather than temporal) white noise pattern. In that example, a decoder of 7 RGCs with knowledge of noise correlations outperforms a decoder of 18 RGCs that assume no noise correlations. The seven RGCs were all included in the 18-cell decoder, and all RGCs had some RF overlap with the decoded pixels. Thus, in its most general formulation, population failure can be considered as a state in which the decoding benefit of using increasingly large populations of neurons is outweighed by the cost of ignoring their noise correlations.

**Light adaptation**. Light adaptation crucially influences how retinal circuits encode visual scenes. Between scotopic and photopic light levels, input to RGCs switches from rod- to cone-mediated pathways. This circuit switch alters both single RGC response properties and correlated activity. For individual RGCs, spatial and temporal integration increases under scotopic conditions[24,25]. Other aspects of RGC activity also depend on light level, including the polarity of stimuli that drive responses, firing rates, and the extent to which spatial integration is linear[39–41]. The switch from rod- to cone-mediated circuits also results in altered common input to RGCs, one of the underlying causes of RGC correlations[11,12,42]. In general, weaker RF surrounds in scotopic conditions result in greater overlap between RF centers and thus more common input between neighboring RGCs. Furthermore, at the low light level used here ($1\,R^{*}\,\mathrm{rod}^{-1}\,\mathrm{s}^{-1}$), AII amacrine cells are expected to be extensively coupled by gap junctions[43], which would also tend to increase the amount of common input between nearby RGCs. Finally, a subset of RGC types are electrically coupled[44], and the strength of this coupling can be altered by light level[45]. Thus, there are several mechanisms by which light adaptation can change both signal and noise correlations among RGCs.

These changes in signal and noise across light levels raise the question of how light adaptation influences information across populations of RGCs. Efficient coding theory—the idea that sensory systems are optimized to encode natural stimuli—has been successful at explaining why RF structure changes across light levels[46–49]. With some notable exceptions[50,51], many applications of this theory assume that RGCs do not exhibit correlated noise, much less that this correlated noise changes with light level. Therefore, a useful direction for future examinations of efficient coding theory is to determine how light-level-dependent changes in correlated activity impact predictions about the optimality of adaptation in RGC responses.

**Implications for downstream processing**. Our results highlight several implications for how downstream circuits may process retinal output. Although the impact of ignoring RGC correlations may depend on particular post-retinal computations, assuming independence among correlated RGCs likely reduces the information that can be extracted from retinal activity. Even in the context of a simple linear readout of RGC responses, ignoring strong noise correlations among RGCs can result in the suboptimal weighting of RGC inputs compared to a weighting determined by an accurate model of correlated noise[52]. In addition, light-level-dependent changes in RGC signals and correlated

noise may place important constraints on post-retinal computations across light levels. For example, downstream circuits that receive input from OFF-bt RGCs may fail to effectively process this input unless they too adapt their processing across light levels. Meanwhile, circuits that receive input from OFF-bs RGCs may be afforded a more static processing strategy. Thus, our results suggest that post-retinal areas may need to differentially process cell-type inputs. Recent work elucidating LGN processing of RGC output confirms that some LGN neurons receive predominant input from a single type of RGC[53–55]. These studies also find LGN neurons with input from diverse types of RGCs, posing further questions of how correlations between RGCs of different types may affect early visual processing.

Studies of light adaptation that span rod-to-cone signaling are relatively common in the retina, but remain sparse in the visual cortex. The few studies that have been performed suggest that V1 RFs are relatively invariant to changes in light level[56]. The insight that RGC responses to the same stimulus change across light levels[39], combined with the fact that correlated noise depends on light adaptation, motivate more research to understand the extent to which V1 and other regions can preserve an invariant representation of visual scenes across light levels.

The importance of accounting for correlated activity is not restricted to retinal processing: in general, ignoring strong noise correlations among any population of neurons will degrade decoding performance, potentially leading to a form of population failure, although the exact effect will depend on the specific downstream computation. While cortical noise correlations may be smaller than those in the retina[57], previous work has demonstrated that ignoring this correlated activity can worsen decoding by 30%[31]. Furthermore, throughout the cortex, brain states has been shown to alter the strength of correlations, such as attentional modulation of correlated noise in V4[58,59]. These findings open the possibility that accounting for correlated noise under certain contexts may be even more important for cortical computations than previously thought. Finally, these principles are likely to be particularly relevant to brain–machine interface applications, where the accuracy of decoding recorded population activity may depend critically on accounting for noise correlations that can change with different brain states and contexts.

## Methods

**MEA recordings**. All experiments were performed in accordance with the guidelines of Duke University's Institutional Animal Care and Use Committee. Long-Evans rats (ages 51–215 days) were dark-adapted overnight and euthanized with an intraperitoneal injection of ketamine/xylazine followed by decapitation. Euthanasia and retinal dissections were performed in darkness with the assistance of infrared converters. We dissected dorsal pieces of the retina that were ~3 × 2 mm large and placed them RGC-side down on an electrode array. The tissue was perfused with oxygenated Ames solution at a rate of 6–8 mL min$^{-1}$. Recordings were performed at 34 °C. The MEA consisted of 512 electrodes with 60-μm spacing, covering an area of 0.9 × 1.8 mm or 519 electrodes with 30-μm spacing, covering a hexagonal area 0.48 μm across[14,60]. The voltage on each electrode was sampled at 20 kHz and filtered between 80 and 2000 Hz.

**Visual stimuli**. Visual stimuli were created with custom Matlab code. Stimuli were presented with a gamma-corrected OLED display (SVGA + XL Rev3, Emagin, Santa Clara, CA). The image from the display was focused onto the photoreceptors using an inverted microscope (Ti-E, Nikon Instruments) with a ×4 objective (CFI Super Fluor ×4, Nikon Instruments). The optimal focus was confirmed by presenting a high spatial resolution checkerboard noise stimulus (20 × 20 μm, refreshing at 15 Hz) and adjusting the focus to maximize the spike rate of RGCs over the MEA. The intensity of the stimulus was set using neutral density filters in the light path. In each recording, stimuli were first presented at the scotopic light level ($1\,Rh^{*}\,\mathrm{rod}^{-1}\,\mathrm{s}^{-1}$), while the retina was in a dark-adapted state. The tissue was adapted to the photopic light level ($10,000\,Rh^{*}\,\mathrm{rod}^{-1}\,\mathrm{s}^{-1}$) for 30 min before continuing recordings at that light level. The refresh rate of the stimulus was 60 Hz and 30 Hz at the photopic and scotopic light levels, respectively. The change in stimulus refresh offset effective contrast changes due to an approximately twofold increase in temporal integration of RGCs from the photopic to scotopic conditions. For GLM fitting and decoding, stimuli consisted of non-repeated, binary white

noise interleaved with repeated, binary white noise segments (5 or 10 s) to control for non-stationarities in recordings. Stimulus pixels in the checkerboard noise were squares with 252 µm sides. These larger stimulus pixels were used because their area was similar to the area of a single RF of the RGC types examined in this study. This size is expected to be near the peak spatial resolution for these RGCs, and they were only ~4× larger than the spatial resolution of the rat visual system (given one cycle/degree resolution and one degree of visual angle spanning 60 µm on the retina)[61]. For cell-type classification, we presented drifting gratings and finer pixel checkerboard noise (63-µm squares, refreshing at 60 Hz)[14].

In a subset of experiments (two retinas), noise correlations were also measured from spontaneous activity and using a natural movie. For the spontaneous activity, spikes were recorded for 30 min with a full-screen gray stimulus. The natural movie was generated by mounting a video camera to the head of a cat walking through a forest[62]. A 10-s segment of this movie was presented 100 times and used to estimate the noise correlations present during naturalistic visual stimulation.

**Spike sorting and neuron identification.** Spikes on each electrode were identified by thresholding the voltage traces at 4 s.d. of a robust-estimate of the voltage s.d. Spike sorting was performed by an automated PCA algorithm and verified by hand with a custom software[63,64]. Spike waveform clusters were identified as neurons only if they exhibited a refractory period (1.5 ms) with <10% estimated contamination. To track identified RGCs across light conditions, cell clusters were sorted in the same PCA subspace at each light level. Neuron identity was verified by checking that EIs and RF locations are stable across conditions[21,22]. RGC types were classified at the photopic light level by first removing direction-selective RGCs, and then clustering using RF properties and autocorrelation shapes[14].

**Measuring noise correlations.** Correlated noise was estimated by subtracting stimulus-driven correlations from the combined signal and noise correlations. Correlations were computed with Matlab's xcov function with normalization to the autocorrelation for each cell in the pair. Response to 100 (or 200) repeats of 10 s (or 5 s) white noise segments, binned at 5 ms, were used to compute CCFs. First, the raw CCF was estimated by averaging the CCFs between two RGCs over all trials. Next, the shuffled CCF (shift predictor)[65] was estimated by using spikes from one repeat for the first cell with spikes from a different repeat for the second cell. The shuffled CCF was averaged over all possible repeat combinations. Subtracting the shuffled CCF from the raw CCF yields the noise CCF. Correlation was quantified with the positive area under each correlogram, the full width of the peak, or the peak height at 0-time lag (negative lobes of the CCFs were not quantified). The spatial scale of correlations for a population was found by fitting the data (e.g., Fig. 1f) to a single-term exponential function. The coefficient of the exponential was the length scale of the correlations.

**GLM fitting.** Open source code was used for GLM fitting (https://github.com/pillowlab/GLMspiketools). GLMs were fit separately at each light level. In total, 50 min of non-repeated white noise was used to fit GLM parameters, with 100 (or 200) 10 s (or 5 s) segments of repeated white noise used for cross-validation. GLM RFs were approximated as rank one[14]: they were composed of the outer product of a spatial filter and temporal filter (which approximated the spatial and temporal RFs, respectively). The temporal filter, spike history filter, and coupling filters were parameterized with a basis of 8 cosine functions. The nonlinearity used was the logexp2 function (available in the GLMspiketools repository); no significant improvement was found using a spline nonlinearity. Only RGCs that had stable responses over the course of the recording were used in the GLM analysis (as judged by a consistent mean firing rate and uniform raster structure to repeated white noise sequences measured early and late in the experiment). For the coupled GLM fits, local populations of RGCs were chosen based on a central RGC that has at least four recorded neighbors. Several RGCs were used in multiple coupled GLMs. Only RGCs with an average firing rate above a threshold were included in the GLM; the threshold was given by the mean minus 1 s.d. of the firing rate of all recorded cells of a certain type. Across light levels, the same groups of RGCs were used to fit GLMs.

**Decoding.** Following the previous work[9], GLM-based decoding was performed by computing the likelihood $p_j$ that a stimulus $x_j$ caused a recorded population response, where $x_j$ is one stimulus option of all possible binary white noise sequences. The decoded estimate comes from Bayes' least squares estimate: $\hat{x} = (\sum p_j x_j)/(\sum p_j)$. We decoded the intensity of one stimulus pixel over six frames in time, or six stimulus pixels in one frame for spatial decoding controls (Supplementary Fig. 4). For the case of decoding one stimulus pixel over time, the decoded pixels were chosen as those that had the greatest weight in the STA of an RGC. Over four retinas, 72 unique pixels were decoded in this study using 169 RGCs. The decoder was provided with the spike times of the RGCs used in the decoding task as well as the intensities of the non-decoded pixels (e.g., see Fig. 3a). This decoding was repeated for 5000 trials on 16 min of non-repeated white noise data (held out from the fitting data). Decoding performance is reported with log SNR calculated from the mutual information between the decoded estimates and presented stimuli[9,26]. Bootstrapped SNRs for error bars were computed for each GLM using 500 subsamples of 3000 trials.

**Simple RF model.** The RF grid model consisted of 25–169 neurons with circular RFs arranged in a hexagonal grid. RF diameters were 250 µm, the same as the stimulus pixel size. Cells' responses to the stimulus had two components: signal plus additive noise. The signals came from filtering the stimulus linearly through the cell's receptive field. When all black pixels cover the receptive field, the signal is zero, and when all white pixels cover it, the signal is 30 Hz. The noise was additive Gaussian noise, with variance equal to the signal (i.e., Poisson variability); this noise was correlated between cells. We varied the magnitude and spatial scale of these correlations in our analyses. The noise correlation strength was set to decrease exponentially with the distance between two cells, where the coefficient of the exponential was the length scale of the correlations.

We discriminated the center pixel's intensity based on the neural responses and the intensities of the non-decoded pixels. This process mimics the GLM decoding, in which we performed the decoding conditioned on the (known) intensities of the non-decoded pixels. To quantify the discriminability, we calculated the quantity $d^2 = \Delta\boldsymbol{\mu}^T Q^{-1} \Delta\boldsymbol{\mu}$[27]. Here, where $\Delta\boldsymbol{\mu}$ is the vector of differences in mean neural firing rates between the conditions where the decoded pixel value was black or white, and differences in the mean intensities of the outer, non-decoded pixels. $Q$ is the mean covariance matrix (averaged over the two conditions: central pixel white or black) of the observables that are used by the decoder (neurons and pixels). This matrix thus consists of four main blocks: one block showing the neuron–neuron covariance, one showing the pixel–pixel covariance, and two showing the neuron-pixel and pixel–neuron covariances, respectively. To quantify the discriminability under the independence assumption, we calculated the quantity

$$d_{\mathrm{d}}^2 = \frac{\left(\Delta\boldsymbol{\mu}^T Q_d^{-1} \Delta\boldsymbol{\mu}\right)^2}{\Delta\boldsymbol{\mu}^T Q_d^{-1} Q Q_d^{-1} \Delta\boldsymbol{\mu}}$$

[27]. Here, $Q_d$ is the covariance matrix under the assumption that the (Poisson) noise is uncorrelated between neurons.

Given a specified set of stimuli (in practice, defined by the mean outer pixel values when the central pixel is white or black, and the pixel–pixel covariance matrices conditioned on the central pixel being white or black), the linear RF model, and the parameters of the neuronal noise correlations, we computed $\Delta\boldsymbol{\mu}$, $Q_d$, and $Q$. We then used those quantities to compute the discriminability when the decoder accounts for correlations ($d^2$) and when it ignores them ($d_{\mathrm{d}}^2$).

For the white noise stimulus (Fig. 7a–e), the mean non-decoded pixel values do not change as the central pixel value changes (so the entries in $\Delta\boldsymbol{\mu}$ corresponding to outer pixel value changes are zero), and the pixel–pixel covariance is a diagonal matrix, reflecting the lack of pixel–pixel correlation. For the natural image stimulus (Fig. 7f–k), we used the ten 512 × 512 pixel grayscale images from Olshausen and Field[66]. We first binarized each image about its median pixel value. Next, 10,000 randomly chosen 5 × 5 pixel patches were used to compute the pixel–pixel covariances and mean outer pixel values, conditioned on the state of the decoded central pixel (white or black). We then averaged these over the ten images and used them in the discriminability calculations.

**Statistics.** Statistical results are found in Table 1 and the text. Because the data generally did not follow a normal distribution, significance tests were performed using the Wilcoxon signed-rank test (for comparisons of one cell type across light levels) and the Wilcoxon rank-sum test (for comparisons across cell types).

**Study design.** The sample size was not predetermined by a statistical method, but our sample sizes (number of recorded cells and number of retinas) are similar to those generally used in the field. Experiments were replicated on multiple retinas, as indicated in figure legends. Data from one retina were excluded because the visual stimulus was not properly focused on the photoreceptors. Randomization and experimenter blinding were not relevant to this study.

**Reporting summary.** Further information on research design is available in the Nature Research Reporting Summary linked to this article.

## Data availability

The data from an example recording generated and analyzed in this study are available at https://doi.org/10.12751/g-node.cnqoty. The full dataset that supports the findings of this study is available from the corresponding author upon reasonable request. Source data are provided with this paper.

## Code availability

The code used to acquire and analyze these data are largely available on public repositories. Stimulus presentation code is available at http://gru.stanford.edu/doku.php/mgl/overview. GLM fitting and decoding code are available at https://github.com/pillowlab/GLMspiketools. Modeling code for Figs. 6 and 7 is available at https://github.com/kmruda/RGC-population-failure.

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

## Acknowledgements

We would like to thank J. Pillow, J. Cafaro, and S. Roy for helpful conversations, and L. Glickfeld, S. Lisberger, and J. Mitchell for reading drafts of the paper. This research was supported by the National Institutes of Health and National Eye Institute grants F31EY028833 (K.R.) and R01EY024567 (G.D.F), the Whitehead Scholars Award (G.D.F), the A.P. Sloan Foundation (J.Z.), Canada Research Chairs program (J.Z.), and the Natural Science and Engineering Research Council of Canada (NSERC; J.Z.).

## Author contributions

K.R. and G.D.F. designed the study. Experiments and analysis were performed by K.R. with assistance from G.D.F. and J.Z. This paper was written by K.R. and G.D.F. with comments from J.Z.

## Competing interests

The authors declare no competing interests.
