## [Peer Review File · Nature Communications]

Reviewers' comments:

Reviewer #1 (Remarks to the Author):

Summary: In this paper the authors investigate how much information is lost by neglecting the correlated variability of neurons when decoding their activity, and in particular how these correlations—and therefore the decoded information—change with environmental conditions such as background luminance. Properly capturing how correlated variability changes with environment conditions how it is used by downstream neural circuitry is an important aspect of neural coding that needs further development. The focus on how lower luminance conditions give result that differ from previous work at higher luminance is the novel contribution of their work. The authors find that the correlated variability of neural activity is even more important at low luminance levels than higher luminance that have been studied in previous work, at least in an OFF-cell type similar to OFF-alpha cells.

Assessment: Overall this was a very well written paper and a solid contribution to furthering our understanding of how neural coding must be adapted to changes in signal and neural variability that change with environment conditions. In going through the paper I often found made notes like “check if authors address X later,” which they inevitably did for almost every note I made; there were only two topics for which I thought discussion was missing, detailed below. I also have a handful of minor comments and questions given below, but I expect they will be easily addressed, and therefore I support publication of the manuscript once discussion of the major points is addressed.

Major comments/questions/suggestions

- The authors only consider OFF cells in the paper, not ON cells or ON-OFF cells. They note that the main OFF cell type (OFF-bt) they studied was chosen so that they could compare to previous experimental results in previous studies (line 80), in particular Nirenberg et al. (2001), Pillow et al. (2008), and Meytlis et al. (2012), and they later they compare to another cell type, OFF-bs. However, Pillow et al. investigated both ON and OFF cells, while Meytlis et al. investigated groups of mixed cell types, with a couple of examples looking only at ON cells and a separate group of ON-OFF cells, so it is not clear to me that the authors' results for OFF cells are directly comparable to some of these cases, nor does it explain why the authors do not consider ON cells in this work. Unless I missed something in these references or I missed an explanation in the paper under consideration, please i) explain somewhere in text why only OFF cells were studied and ii) when comparing the increases that those studies saw to the authors' work add a caveat that they did not use the same cell types (e.g., around line 429 in the discussion). Given the authors' observation that the effect of light modulation was much stronger in one of the OFF cell types they studied, it may be worth pointing out the fact that the changes in information the authors observed in bright light conditions are consistent with these changes observed for other cell types.
- The stimuli the authors used is a binary white-noise checkerboard stimulus, as opposed to more naturalistic stimuli. This is common and a perfectly suitable approach, but I was expecting the discussion to address how more naturalistic stimuli might change their results, for several possible reasons, including the difficulty of GLMs to reliably capture natural scenes—see, e.g., Heitman et al. (2016), cited but not in the context of natural scenes—or because of the statistical structure of the scenes themselves. Notably, Meytlis et al. (2012) found that correlations did not seem to be as important for natural scenes as white noise stimuli, so it would be particularly interesting to address whether this might be expected to hold at lower light levels, or whether it might be a failure of decoding from GLM models fits to natural scene statistics.

To be very clear, I am not asking the authors to perform any new analyses to answer these questions, just to add some discussion of the issues and caveats in the text.

Minor comments/questions /suggestions:

- I am a bit unclear on the definition of the cross-correlation functions. Which signals, exactly, are being used to compute the CCFs? The units are given as firing rate (Hz) in Fig. 1. If the cross-correlation is between firing rates, should the units not be firing rate squared? If the signals are the raw spike trains, wouldn't the units be spikes squared? The exact mathematical definition used to compute these functions is not written out, only described in the methods, and it does not explicitly say which signals are used. Please clarify.
- P. 7, lines 183-184. The procedure describes using local clusters of neurons to perform the analysis (and the authors later check how much this would be impacted by including neurons that are further away). An additional justification the authors could optionally mention for why one might expect that looking only at local clusters suffices is that while in principle downstream circuitry could receive convergent inputs from nonlocal clusters, the retinotopic topology of the visual system suggests that the downstream circuitry maintains this locality.
- P. 8, line 198 (& other places in which percent-increases in information are reported): Please clarify in text on first usage of this measure which definition of "information" is used being used. Searching through the methods I eventually found that it is referring to mutual information as estimated by log SNR, but I originally thought it might be referring to Fisher Information (or precision) and not mutual information.
- Fig. 7E: Rainbow color-scales are not recommended as the intensities of the colors are not monotonic and therefore become ambiguous when printed in greyscale. Something like Matlab's `parula` would be a better colormap (or some other choice with monotonic intensities).
- P. 15, lines 487 and following, in particular lines 490-492. The wording of this paragraph comes off as suggesting that the efficient coding framework as a whole assumes that noise is not correlated, rather than the first applications of the idea. I suggest a slight rewording, "Efficient coding theory—the idea that sensory systems are optimized to encode natural stimuli—has been successful at explaining why RF structure changes across light levels (Attneave, 1954; Barlow, 1961; Atick and Redlich, 1990; Van Hateren, 1993). However, many applications of this theory to retinal coding assume that RGCs do not exhibit correlated noise, much less that this correlated noise changes with light level."
 - o A couple of citations that address some aspects of noise correlation effects on neural coding that may optionally be worth adding here are "Sensory noise predicts divisive reshaping of receptive fields" by Chalk et al. (<https://doi.org/10.1371/journal.pcbi.1005582>), which looks at how sensory noise reshapes RFs, and addresses noise correlations in the SI, and "How Do Efficient Coding Strategies Depend on Origins of Noise in Neural Circuits?" by Brinkman et al. (<https://doi.org/10.1371/journal.pcbi.1005150>), which model how noise correlations affect efficient coding in pairs of neurons, although the model neurons do not have RFs.
- P.18, line 591. Does "positive area" mean that negative lobes of the CCF are ignored, or that the absolute value of the CCF is taken as the measure of correlation? Please clarify.
- P. 18, line 599. Regarding the rank 1 approximation of the receptive fields, I would cite here the reference that these RFs are space-time separable that was given in the Supplementary Figure 2 caption.
- P. 19, line 629 & surrounding. I am not sure what the statement "To match the GLM decoding, here the covariance includes covariance of neurons and outer pixel intensities" means. It sounds to me like the covariance matrix Q includes not just the neural covariances, but also covariances of some pixel intensities, though I'm not sure which pixels are "outer" pixels—the ones near the edge of the RF

overlap? Does Q contain neuron-outer pixel covariances? In this case $\Delta \mu$ will also contain mean pixel intensities? If so, it is not mentioned in text. Please clarify this paragraph, as it does not give me a clear picture of the model or the calculation of d.

Best,
Braden Brinkman

Reviewer #2 (Remarks to the Author):

The manuscript by Ruda, Zylberberg, and Field explores the role noise correlations on the population code of retinal ganglion cells (RGCs) in rat whose spikes were measured on a multi-electrode-array (MeA). This is an important topic, and overall the writing is remarkably elegant and precise. The data and analyses are of very high quality, and the Methods section is comprehensive in covering experimental, analysis, and modeling methods. The authors do an excellent job setting up the problem and explaining the results both mathematically and with intuition for what they mean. There are a number of well-founded and significant contributions in this work (along with several more minor results that largely confirm previous work):

1. Noise correlations differ substantially based on light level in OFF bT RGCs.
2. Noise correlations are much smaller among OFF bS RGCs.
3. Assuming independent noise in decoding the temporal or spatial structure of a particular large-checker checkerboard stimulus causes very large decoding errors when the correlations are large.

Points 1 and 2 are demonstrated clearly in the data, so I will not comment on them further. Point 3 is the focus of the majority of the manuscript, and this is where I have some concerns about interpretation and generality.

Major:

How dependent are the results on the stimulus? While the authors do an excellent job explaining and controlling for assumptions of their models, there is no such control for the selection of the particular checkerboard stimulus. And to be clear, this is an unusual stimulus. Unlike the high-resolution checkerboards used to measure spatiotemporal RFs (like the $20 \mu\text{m} \times 20 \mu\text{m}$ checkerboard used in this study for RF measurement), the stimulus used for the correlation and decoding studies had a checker size of $240 \mu\text{m}$. This is around the 2-sigma RF size of these RGCs, so most checkers will predominantly activate one to three RGCs strongly and maybe a few more surrounding ones weakly. This is roughly the spatial scale of the correlations (Fig. 1E and Fig. S1), and roughly the relevant population size for the GLMs (at least for the temporal decoding). The reliance on this stimulus raises two questions.

1. Are noise correlations the same under different stimulus conditions? I understand that the authors performed the correct mean subtraction to eliminate stimulus correlations in the analysis, but that does not mean that noise correlations themselves would be the same under different stimulus conditions. In fact, the present work demonstrates that noise correlations depend dramatically on light level (point 1 above). If they also depend substantially on the stimulus, there is a danger that the results become highly restricted to this particular stimulus.
2. Even if the authors were to measure noise correlations in different stimulus conditions (which I suggest they should do) and show that noise correlations don't depend substantially on stimulus condition at a given light level, there is still an issue with their interpretation. The brain does not have the luxury of separating stimulus from noise correlations on single trials by subtracting the mean and

computing correlations on the residuals. This raises several related questions:

A. How large are noise correlations compared to the stimulus correlations in typical natural scenes?

B. Is the assumption of independence more problematic in its ignorance of stimulus correlations than in its ignorance of noise correlations?

I realize that stimulation with natural scenes introduces a host of complications for analysis – not the least of which is quantifying a decoding metric. Perhaps there is something to be done with pink noise, which would capture the spatial frequency distribution of natural scenes but not their phase composition. One way or another, the authors must address the generality of these findings to more naturalistic stimuli and the interaction between stimulus and noise correlations.

Moderate:

Maybe this is more stylistic than substantive, but I think the comparison to single cells and the term “population failure” are a bit over-emphasized. It is indeed an interesting point that the failure of assuming independence can overwhelm the benefit of adding more cells to a decoder, but what is so special about this one vs. many comparison? It makes intuitive sense that adding what amounts to garbage information from surrounding cells because you have a very poor noise model could be worse than ignoring those cells completely. Calling population failure a “mode” seems to impart a false dichotomy on the situation. Instead, as the authors show, the reduction in information your decoder suffers by assuming the incorrect noise model varies smoothly as a function of the size and extent of the noise correlations. The fact that it becomes worse than a single cell decoder just means that it crossed a particular threshold – and this is very likely to depend on how much information single cells carry in the particular decoding task. Since the checkerboard squares were about the size of a whole RF center, it stands to reason that a single-cell decoder would perform at least decently well on the task. For a different task, like encoding the position of a spot, it may be impossible for a single-cell decoder to outperform a population code for the trivial reason of the limited size of a single RF.

Minor:

A study from Michael Berry’s lab looked at decoding spatial structure from populations of RGCs and found a related result in that codes that assumed independence were incapable of reaching low error rates (Schwartz et al., 2012). The stimuli were different, and the focus on that work was on larger populations and on stimulus correlations. Still, it is worth discussing how the present results relate to that study.

Reference

Schwartz G, Macke J, Amodè D, Tang H, Berry MJ (2012) Low error discrimination using a correlated population code. *J Neurophysiol* 108:1069–1088 Available at: <http://www.pubmedcentral.nih.gov/articlerender.fcgi?artid=3424080&tool=pmcentrez&rendertype=abstract>.

Reviewer #3 (Remarks to the Author):

Ruda, Zylberberg and Field present an analysis of the impact of noise correlation in the activity of rat retinal ganglion cells across photopic and scotopic conditions. They combined multi-electrode array experiments with a computational analysis of the decoding performance based on GLM modeling. From abstract, result and discussion, the main results of the paper are two: (i) OFF-bt RGCs increase their noise-correlation in scotopic condition and this induces a large improvement in the decoding performance when noise-correlations are taken into account. (ii) assuming independent cells cause a drop in performance when decoding. There is a whole literature on how noise-correlations impact information transmission in neuronal systems, and in particular in the retina. This paper brings a

contribution by focusing on the decoding, instead of the encoding. This point of view is not new, but of equal, if not larger, importance and surely less investigated than the other.

From a pure theoretical point of view, point (ii) is not particularly interesting. The fact that by decoding with a wrong model (independent GLM, figs 4&5 or independent linear model, figs 6&7) you get lower performance is not surprising. What is interesting is that the decrease of performance can be very large and is even larger than decoding from a single cell. The authors then construct two toy models and show that this effect depends on the ranges of parameters. For some ranges the difference is minimal, for others the improvement can be very large (population failure). So the question here is where the real retina lies in this scenario. This is where it comes point (i): when applied to real data, and especially from scotopic regimes, the authors found a strong population failure (Fig 2), but only for OFF-bt cells and not for OFF-bs (Fig.3).

On a side note, and as the authors state in the discussion, the fact that noise correlations increase in scotopic regimes is not new and well documented in the literature. Similarly their results in the improvement of information transmission in photopic condition are consistent with previous findings.

I found both points (i) and (ii) interesting, but I don't think the obtained results are much novel and/or convincing to support publication, at least not in the present form, and I believe they should be developed further. In particular the role of the stimulus is quite absent in the author analysis, and this is a major lack when dealing with an early sensory system like the retina. More details in the following. Also, the fact that the impact on decoding performance by neglecting noise-correlations is very large in scotopic condition is interesting but it is also a direct consequence of the larger noise correlations in this regime.

Overall the paper is well written and easy to read. Good work here. Given the interdisciplinary topic here addressed, this was not easy.

Major points

Toy model of Fig. 6, two neurons with gaussian noise responding to two stimuli. In this simple case most of the computation can be performed analytically using the fisher information and the d-prime approach (Seung and Sompolinsky 1993). Previous works in the literature have shown that the increase of information transmission or decoding depends on the stimulus, and in particular if stimulus and noise correlations have the same or opposite sign (differential-correlations in Moreno-Bote et al Nature 2014, see also Franke et al Neuron 2016). The fact that the authors find a signature of population failure here is interesting, but I would require a full analysis of this simple model upon changing the strength of noise correlations, and the extent and sign of stimulus correlation. As the authors found, noise correlations are positive in the retina, but signal correlations can be both positive or negative depending on the stimulus. Strictly speaking, in your example both neurons have a larger firing rate for stimulus 2, what if only one does, but the other decreases its mean rate?

From this simple drawing, it seems to me that the population failure should be much weaker, if not absent at all if the neurons are negatively stimulus-correlated. I think that the authors should take advantage of the possibility to perform all the computations analytically and present a complete picture on when and with what extent population failure happens.

The lack of the analysis of the role of stimulus is even more evident on the retinal results of Figs 4&5. The authors present retinal responses to checkerboard stimulation with checks of a rather large side of 240 microns. This stimulus induces strong stimulus correlation between nearby cells. Therefore it allows for analysing only a rather small spectrum of possible combinations of stimulus and noise correlations. In my opinion this limitation weakens strongly the validity of the authors conclusion,

namely the fact that population failure is a relevant phenomenon for downstream neurons. In conclusion, I found this result not satisfying, and I would recommend enriching it by the analysis of other stimuli that can induce both positive and negative stimulus correlations, as for example drifting gabors.

In the paper they are missing, but often multi-electrode array experiments also record ON cells. Do they show noise-correlations?

Also, do you find noise-correlations also between cells of different types?

Minor points

Figure 1. Even if partially shown in figure 2, it would be interesting here to see the 6 PSTHs of cells i,ii,iii in the two conditions.

What is the time-bin length for computing the cross-correlation of panel D? If it is of the order of 10ms, It would be nice to see, maybe in an inset, the peak of the correlations at finer time-scale (1-2ms).

How much of the changes in the cross-correlation between photopic and scotopic conditions are due to changes in the auto-correlation of individual neurons?

Figure 2. Panel D, why did you show the data from only one retina?

Panel F. it seems that you over-estimate the CCF peak in photopic condition, but it is hard to see it clearly. Could you split this panel in two, one panel for each condition please?

How much differ the GLM filters in the two conditions? Can you compare them please?

Decoding. I found that some important details are missing on how the decoding model is constructed. The author explains that they use optimal bayesian decoding to predict stimulus value in one pixel for 6 temporal windows. However they do not explain clearly what are the input of the decoder. From figure 3, I understood that it is fed with the spiking times of the small group of cells, but what about the other stimulus pixels around the central one? Do you need them?

Also, it is not clear to me if you decoded the stimulus only in cases where a central cell has a large overlap with the pixel. If it is the case, how many pixels over the total were you able to decode?

Figure 4. I have hard times in understanding what is plotted in panel D

Figure 6. Following the first major point, I suggest adding an example of negatively stimulus-correlated neurons (example the toy picture above).

Limits of x-axis in panel A&B and D&E should be the same for readability

Figure 7. Could you add panels also for the single cell decoder please?

Supp fig 2. Panel C&D. Left, why does the independent GLM have higher performance than the coupled one?

Right. Is this the same metric from Fig.2D? If not, it should be better clarified.

Panel F, as in Fig2, this is hard to see, you should split the two conditions.

L191-192 To which paragraph the "see methods" refers to?

Editor's Note:

Your manuscript entitled "Ignoring correlated activity causes a failure of retinal population codes" has now been seen by three referees, whose comments are appended below. You will see from their comments copied below that while they find your work of considerable potential interest, they have raised quite substantial concerns that must be addressed. In light of these comments, we cannot accept the manuscript for publication, but would be interested in considering a revised version that addresses these serious concerns, in particular those regarding the noise correlations and the stimulus size.

We hope you will find the referees' comments useful as you decide how to proceed. Should further experimental data or analysis allow you to address these criticisms, we would be happy to look at a substantially revised manuscript. However, please bear in mind that we will be reluctant to approach the referees again in the absence of major revisions. If the revision process takes significantly longer than three months, we will be happy to reconsider your paper at a later date, as long as nothing similar has been accepted for publication at Nature Communications or published elsewhere in the meantime.

General Response:

The primary comment expressed by all three reviewers was to show more evidence that our results generalize to more natural stimulus conditions. Specifically, the Reviewers asked (1) does the amount of noise correlations between RGCs depend on the stimulus and (2) does population failure depend on type of decoded stimulus (e.g. checkerboard noise versus a natural scene)?

To answer the first of these questions, we have included new experiments and data that illustrate how the strength of the noise correlations depends on the stimulus. Specifically, we compare noise correlations during spontaneous activity with those measured during the presentation of (1) large checkerboard noise, (2) small checkerboard noise, and (3) a natural movie (Supp. Fig. 1). We performed this comparison at both the rod and cone light levels. Across stimulus conditions, the noise correlations were similar. However, the noise correlations were somewhat higher under spontaneous activity, small checkerboard noise and the natural movie, than during the large checkerboard noise. Thus, our stimulus provides a conservative estimate on the strength of the noise correlations (Supp. Fig. 1B and D). Given our results in Figs. 6, 7, Supp. Fig. 5, and Supp. Fig. 7, these conservative estimates on the noise correlations will produce conservative estimates on the magnitude and frequency of population failure. We think the estimated noise correlations are somewhat smaller during the large-checkerboard noise because this stimulus drives strong responses in the neurons which pushes them toward saturation, thereby reducing the dynamic range upon which the noise correlations can act. Similar effects have been observed previously (see Fig. 5 of Greschner et al 2011, J Neurophys).

To answer the second question (does population failure depend on the decoded stimulus?), we have utilized the model RGC array presented in Fig. 7 to discriminate pixels within natural scenes. As a reminder, the analysis in Fig. 7 presents a checkerboard stimulus to a population of modeled RGCs whose responses on each simulated trial were given by: a) the overlap between their receptive fields and the bright pixels in the stimulus plus b) Gaussian (pseudo-Poisson) noise, which was potentially correlated between neurons. We then modified the noise correlations between these neurons, and studied how well the stimulus could be decoded from the neural responses. In our new analyses, we utilize the same model to discriminate pixels in natural scenes (see new Supp. Fig. 7). We show that 'population failure' occurs under very similar amounts of correlated noise when discriminating natural scenes versus checkerboard patterns (Supp. Fig. 7 compared with Fig. 7). These new analyses also support the conclusion that our estimates of the strength of noise correlations provide a

conservative estimate on the importance of accounting for noise correlations when decoding (see preceding paragraph). This is because stronger noise correlations produce more severe population failure regardless of whether the decoded stimulus is a checkerboard pattern or a natural scene.

These tests indicate that the key findings in this manuscript generalize well to more natural stimulus conditions. Thus, we think we have addressed the core comments/concerns. We respond to each Reviewer's specific comments below.

Reviewers' comments:

Reviewer #1 (Remarks to the Author):

Summary: In this paper the authors investigate how much information is lost by neglecting the correlated variability of neurons when decoding their activity, and in particular how these correlations—and therefore the decoded information—change with environmental conditions such as background luminance. Properly capturing how correlated variability changes with environment conditions and how it is used by downstream neural circuitry is an important aspect of neural coding that needs further development. The focus on how lower luminance conditions give results that differ from previous work at higher luminance is the novel contribution of their work. The authors find that the correlated variability of neural activity is even more important at low luminance levels than higher luminance that have been studied in previous work, at least in an OFF-cell type similar to OFF-alpha cells.

Assessment: Overall this was a very well written paper and a solid contribution to furthering our understanding of how neural coding must be adapted to changes in signal and neural variability that change with environment conditions. In going through the paper I often made notes like “check if authors address X later,” which they inevitably did for almost every note I made; there were only two topics for which I thought discussion was missing, detailed below. I also have a handful of minor comments and questions given below, but I expect they will be easily addressed, and therefore I support publication of the manuscript once discussion of the major points is addressed.

Major comments/questions/suggestions

- The authors only consider OFF cells in the paper, not ON cells or ON-OFF cells. They note that the main OFF cell type (OFF-bt) they studied was chosen so that they could compare to previous experimental results in previous studies (line 80), in particular Nirenberg et al. (2001), Pillow et al. (2008), and Meytlis et al. (2012), and they later they compare to another cell type, OFF-bs. However, Pillow et al. investigated both ON and OFF cells, while Meytlis et al. investigated groups of mixed cell types, with a couple of examples looking only at ON cells and a separate group of ON-OFF cells, so it is not clear to me that the authors' results for OFF cells are directly comparable to some of these cases, nor does it explain why the authors do not consider ON cells in this work. Unless I missed something in these references or I missed an explanation in the paper under consideration, please i) explain somewhere in text why only OFF cells were studied and ii) when comparing the increases that those studies saw to the authors' work add a caveat that they did not use the same cell types (e.g., around line 429 in the discussion). Given the authors' observation that the effect of light modulation was much stronger in one of the OFF cell types they studied, it may be worth pointing out the fact that the changes in information the authors observed in bright light conditions are consistent with these changes observed for other cell types.

We thank the Reviewer for this comment and helpful suggestion to better explain some of our analysis choices and their relationship to previous work. We have added a paragraph to the first subsection of the Discussion to explain our choice for focusing on OFF RGCs. This choice was based on their mosaics being the most complete and their spikes being well isolated. That paragraph begins on line 380. We have also added a phrase on line 373 to make it clear that the cell types and species used across our study, the Pillow et al 2008 (Nature) and the Meytlis et al 2012 (Vision Res) studies are distinct.

- The stimuli the authors used is a binary white-noise checkerboard stimulus, as opposed to more naturalistic stimuli. This is common and a perfectly suitable approach, but I was expecting the discussion to address how more naturalistic stimuli might change their results, for several possible reasons, including the difficulty of GLMs to reliably capture natural scenes—see, e.g., Heitman et al. (2016), cited but not in the context of natural scenes—or because of the statistical structure of the scenes themselves. Notably, Meytlis et al. (2012) found that correlations did not seem to be as important for natural scenes as white noise stimuli, so it would be particularly interesting to address whether this might be expected to hold at lower light levels, or whether it might be a failure of decoding from GLM models fits to natural scene statistics. To be very clear, I am not asking the authors to perform any new analyses to answer these questions, just to add some discussion of the issues and caveats in the text.

We agree with the Reviewer that the extent to which our results generalize to conditions that are more natural is important to address. A key way in which our results would fail to generalize would be if noise correlations depended strongly on the nature of the stimulus. For example, perhaps noise correlations become very weak in the presence of natural visual stimuli. Given our theoretical analyses presented in Figs. 6 and 7, this would indicate a much smaller cost for ignoring those correlations and would substantially decrease the frequency of 'population failure'.

Thus, we have included new experiments and data that illustrate how the strength of the noise correlations depends on the stimulus. Specifically, we compare noise correlations during spontaneous activity with those measured during the presentation of (1) large checkerboard noise, (2) small checkerboard noise, and (3) a natural movie (Supp. Fig. 1). The natural movie consisted of a camera mounted to the head of a cat walking through a forest (Betsch et al 2004, Biol Cybern). We performed this comparison at both the rod and cone light levels. Across stimulus conditions, the noise correlations were similar. However, the noise correlations were somewhat higher under spontaneous activity, small checkerboard noise and the natural movie, than during the large checkerboard noise. Thus, our stimulus provides a **conservative estimate** on the strength of the noise correlations (Supp. Fig. 1B and D). Given our results in Figs. 6, 7, Supp. Fig. 5, and Supp. Fig. 7, these conservative estimates on the noise correlations will produce conservative estimates on the magnitude and frequency of population failure. We think the estimated noise correlations are somewhat smaller during the large-checkerboard noise because this stimulus drives strong responses in the neurons which pushes them toward saturation, thereby reducing the dynamic range upon which the noise correlations can act. Similar effects have been observed previously (see Fig. 5 of Greschner et al 2011, J Neurophys).

To summarize, under more natural stimulus conditions (a natural movie) noise correlations are even a little stronger than our estimates using large checkerboard noise. Thus our estimates of the noise correlations likely represent a lower bound on their strength during natural vision. We discuss the implications of these observations in the Discussion, lines 420.

Also, to address comments by all of the Reviewers, we examined the effect of noise correlations on discriminating pixels in natural scenes using the model presented in Fig. 7. As a reminder, the analysis in Fig. 7 presents a checkerboard stimulus to a population of modeled RGCs whose responses on each simulated trial were given by: a) the overlap between their receptive fields and the bright pixels in the stimulus plus b) Gaussian (pseudo-Poisson) noise, which was potentially correlated between neurons. We then modified the noise correlations between these neurons, and studied how well the stimulus could be decoded from the neural responses. In our new analysis (Supp. Fig. 7 of the revision), we used binarized natural image stimuli. This analysis yielded very similar trends in population failure as when using checkerboard noise, indicating there is nothing specific or particular about the population failure results for checkerboard noise stimuli, and that these results generalize to more natural stimulus conditions.

Finally, we comment on line 435 of the manuscript for why these results may differ from those in the Meytlis et al 2012 (Vision Res) study.

Minor comments/questions /suggestions:

- I am a bit unclear on the definition of the cross-correlation functions. Which signals, exactly, are being used to compute the CCFs? The units are given as firing rate (Hz) in Fig. 1. If the cross-correlation is between firing rates, should the units not be firing rate squared? If the signals are the raw spike trains, wouldn't the units be spikes squared? The exact mathematical definition used to compute these functions is not written out, only described in the methods, and it does not explicitly say which signals are used. Please clarify.

To avoid confusion, we now plot CCFs in units of correlation (-1 to 1) rather than firing rate. CCFs were computed using the `xcov` function in Matlab, with a normalization to the autocorrelation of both neurons (the 'coef' option). The procedure for computing correlation is described in the Methods beginning on line 574.

- P. 7, lines 183-184. The procedure describes using local clusters of neurons to perform the analysis (and the authors later check how much this would be impacted by including neurons that are further away). An additional justification the authors could optionally mention for why one might expect that looking only at local clusters suffices is that while in principle downstream circuitry could receive convergent inputs from nonlocal clusters, the retinotopic topology of the visual system suggests that the downstream circuitry maintains this locality.

While in general we agree with the Reviewer, given observations of 'extra-classical' surrounds for V1 receptive fields being spatially extended and strongly modulatory, we have decided to forego this additional justification.

- P. 8, line 198 (& other places in which percent-increases in information are reported): Please clarify in text on first usage of this measure which definition of "information" is being used. Searching through the methods I eventually found that it is referring to mutual information as estimated by log SNR, but I originally thought it might be referring to Fisher Information (or precision) and not mutual information.

We have made it clear on line 173 we are computing mutual information.

"We report decoding performance with a signal-to-noise ratio (SNR), which quantifies the mutual information rate in bits/s that the decoded estimate provides about the actual stimulus (Warland et al., 1997)."

- Fig. 7E: Rainbow color-scales are not recommended as the intensities of the colors are not monotonic and therefore become ambiguous when printed in greyscale. Something like Matlab's parula would be a better colormap (or some other choice with monotonic intensities).

We agree and have changed the colormap to parula.

P. 15, lines 487 and following, in particular lines 490-492. The wording of this paragraph comes off as suggesting that the efficient coding framework as a whole assumes that noise is not correlated, rather than the first applications of the idea. I suggest a slight rewording, "Efficient coding theory—the idea that sensory systems are optimized to encode natural stimuli—has been successful at explaining why RF structure changes across light levels (Attneave, 1954; Barlow, 1961; Atick and Redlich, 1990; Van Hateren, 1993). However, many applications of this theory to retinal coding assume that RGCs do not exhibit correlated noise, much less that this correlated noise changes with light level."

We thank the reviewer for helping us avoid this somewhat clumsy phrasing. We have changed this sentence as suggested (see line 469).

A couple of citations that address some aspects of noise correlation effects on neural coding that may optionally be worth adding here are "Sensory noise predicts divisive reshaping of receptive fields" by Chalk et al. (<https://doi.org/10.1371/journal.pcbi.1005582>), which looks at how sensory noise reshapes RFs, and addresses noise correlations in the SI, and "How Do Efficient Coding Strategies Depend on Origins of Noise in Neural Circuits?" by Brinkman et al. (<https://doi.org/10.1371/journal.pcbi.1005150>), which model how noise correlations affect efficient coding in pairs of neurons, although the model neurons do not have RFs.

Thank you, we have added the citations.

"With some notable exceptions (Brinkmann2016, Chalk2017), many applications of this theory assume that RGCs do not exhibit correlated noise, much less that this correlated noise changes with light level."

- P.18, line 591. Does "positive area" mean that negative lobes of the CCF are ignored, or that the absolute value of the CCF is taken as the measure of correlation? Please clarify.

Yes, we computed CCF area by ignoring the negative lobes. This is clarified on line 583.

- P. 18, line 599. Regarding the rank 1 approximation of the receptive fields, I would cite here the reference that these RFs are space-time separable that was given in the Supplementary Figure 2 caption.

We have added the reference to Ravi et al 2018.

- P. 19, line 629 & surrounding. I am not sure what the statement "To match the GLM decoding, here the covariance includes covariance of neurons and outer pixel intensities" means. It sounds to me like the covariance matrix Q includes not just the neural covariances, but also covariances of some pixel intensities, though I'm not sure which pixels are "outer" pixels—the ones near the edge of the RF overlap? Does Q contain neuron-outer pixel covariances? In this case $\Delta \mu$ will also contain mean pixel intensities? If so, it is not mentioned in text. Please clarify this paragraph, as it does not give me a clear picture of the model or the calculation of d.

We thank the reviewer for pointing out this needed clarification. The reviewer is correct that $\Delta \mu$ includes mean neural firing rates and pixel intensities, and the covariance matrix Q includes neural covariances, pixel covariances, and neuron-pixel covariances. We refer to “outer” pixels as all pixels other than the central pixel that was decoded. That central pixel is not included in Q or $\Delta \mu$, as the decoding task is to discriminate its intensity. We have clarified this in the “Simple RF Model” section beginning on Line 617 to 650.

Best,
Braden Brinkman

Reviewer #2 (Remarks to the Author):

The manuscript by Ruda, Zylberberg, and Field explores the role noise correlations on the population code of retinal ganglion cells (RGCs) in rat whose spikes were measured on a multi-electrode-array (MeA). This is an important topic, and overall the writing is remarkably elegant and precise. The data and analyses are of very high quality, and the Methods section is comprehensive in covering experimental, analysis, and modeling methods. The authors do an excellent job setting up the problem and explaining the results both mathematically and with intuition for what they mean. There are a number of well-founded and significant contributions in this work (along with several more minor results that largely confirm previous work):

1. Noise correlations differ substantially based on light level in OFF bT RGCs.
2. Noise correlations are much smaller among OFF bS RGCs.
3. Assuming independent noise in decoding the temporal or spatial structure of a particular large-checker checkerboard stimulus causes very large decoding errors when the correlations are large.

Points 1 and 2 are demonstrated clearly in the data, so I will not comment on them further. Point 3 is the focus of the majority of the manuscript, and this is where I have some concerns about interpretation and generality.

We thank the reviewer for their clear summary, highlighting the strengths of the manuscript, and their constructive feedback.

Major:

How dependent are the results on the stimulus? While the authors do an excellent job explaining and controlling for assumptions of their models, there is no such control for the selection of the particular checkerboard stimulus. And to be clear, this is an unusual stimulus. Unlike the high-resolution checkerboards used to measure spatiotemporal RFs (like the $20 \mu\text{m} \times 20 \mu\text{m}$ checkerboard used in this study for RF measurement), the stimulus used for the correlation and decoding studies had a checker size of $240 \mu\text{m}$. This is around the 2-sigma RF size of these RGCs, so most checkers will predominantly activate one to three RGCs strongly and maybe a few more surrounding ones weakly. This is roughly the spatial scale of the correlations (Fig. 1E and Fig. S1), and roughly the relevant population size for the GLMs (at least for the temporal decoding). The reliance on this stimulus raises two questions.

1. Are noise correlations the same under different stimulus conditions? I understand that the authors performed the correct mean subtraction to eliminate stimulus correlations in the analysis, but that does not mean that noise correlations themselves would be the same under different stimulus conditions. In fact, the present work demonstrates that noise correlations depend dramatically on light level (point 1 above). If they also depend substantially on the stimulus, there is a danger that the results become highly restricted to this particular stimulus.

We agree with the reviewer that investigating the extent to which the measured noise correlations extend to other stimulus conditions is important. Previous work in the primate retina (Fig. 5 of Greschner et al, 2011 J Neurophys.) showed that the amplitude of noise correlations depended very weakly on the spatial scale of correlations in a checkerboard stimulus. Nevertheless, we have included in the revised manuscript three comparisons.

Specifically, we compare noise correlations during spontaneous activity with those measured during the presentation of (1) large checkerboard noise, (2) small checkerboard noise, and (3) a natural movie (Supp. Fig. 1). The natural movie consisted of a camera mounted to the head of a cat walking through a forest (Betsch et al 2004, Biol Cybern). We performed this comparison at both the rod and cone light levels. Across stimulus conditions, the noise correlations were similar. However, the noise correlations were somewhat higher under spontaneous activity, small checkerboard noise and the natural movie, than during the large checkerboard noise. Thus, our stimulus provides a conservative estimate on the strength of the noise correlations (Supp. Fig. 1B and D). Given our results in Figs. 6, 7, Supp. Fig. 5, and Supp. Fig. 7, these conservative estimates on the noise correlations will produce conservative estimates on the magnitude and frequency of population failure. We think the estimated noise correlations are smaller during the large-checkerboard noise because this stimulus drives strong responses in the neurons which pushes them toward saturation, thereby reducing the dynamic range upon which the noise correlations can act. Similar effects have been observed previously (see Fig. 5 of Greschner et al 2011, J Neurophys).

To summarize, under more natural stimulus conditions (e.g. a natural movie) noise correlations are even a little stronger than our estimates using large checkerboard noise (see Supp. Figs. B & E). Thus our estimates of the noise correlations likely represent a lower bound on their strength during natural vision. We discuss the implications of these observations in the Discussion, lines 420-448.

The Reviewer also comments that the checkers we used are particularly large. We point out that the area of each checker is actually similar to the area of the receptive field center for the RGCs that we are using to decode (See Figs. 3A, 4A, and 5A for examples of RFs and the pixel sizes that were decoded). The RFs had an average diameter of 259 μm (at 1-sigma) and the pixels had an area of 252 μm^2 on each edge. Second, the size of these checkers is only ~4-fold higher than the spatial resolving acuity of the rat visual system, which is 1 cycle/deg (Prusky et al., 2000, Vision Research), which corresponds to ~60 $\mu\text{m}/\text{deg}$ on the retina. The linear size of the checkers is 252 μm , which is 60 x 4.2. Thus, we think the parameters of this stimulus we used are a reasonable match to the RFs sizes we are examining and the acuity of the rat's visual system.

2. Even if the authors were to measure noise correlations in different stimulus conditions (which I suggest they should do) and show that noise correlations don't depend substantially on stimulus condition at a given light level, there is still an issue with their interpretation. The brain does not have the luxury of separating stimulus from noise correlations on single trials by subtracting the mean and computing correlations on the residuals. This raises several related questions:

A. How large are noise correlations compared to the stimulus correlations in typical natural scenes?

B. Is the assumption of independence more problematic in its ignorance of stimulus correlations than in its ignorance of noise correlations?

We think these comments may reflect a misunderstanding about what the models we compare are preserving and eliminating in the signals (and noise) among the RGCs. The independent, uncoupled GLM preserves signal correlations (correlations induced by the stimulus); it only eliminates noise correlations. So the diminished performance that we observe in the independent GLM is specifically attributable to assuming the neurons are noise independent because the stimulus-induced correlations are the same between the coupled GLM and the independent GLM. We have added several lines to the manuscript to make this point clear (See e.g. beginning at line 157), and we are now more careful throughout the manuscript to clarify when we are referring to noise versus signal correlations.

This is also part of the reason why we think the observation of ‘population failure’ is so interesting. The single cell decoder doesn’t know anything about the signal correlations across RGCs or their noise correlations, yet it can outperform a model that knows about the structure of the signal correlations (but assumes the noise is independent). This indicates that failure to account for noise correlations in decoding can and often does outweigh the benefit of knowing added signals from other neurons, along with their signal correlations. We also note that Fig. 7 examines the relationship between signal and noise correlations, where changing the amount of receptive field overlap in the model is a way to manipulate the amount of signal correlations (more RF overlap results in higher signal correlations).

We hope these clarifications address the Reviewer’s concern.

I realize that stimulation with natural scenes introduces a host of complications for analysis – not the least of which is quantifying a decoding metric. Perhaps there is something to be done with pink noise, which would capture the spatial frequency distribution of natural scenes but not their phase composition. One way or another, the authors must address the generality of these findings to more naturalistic stimuli and the interaction between stimulus and noise correlations.

We think the new version of the manuscript addresses this concern in several ways. First, as stated above, the nature of our analysis already isolates noise correlations: signal correlations are preserved in the independent GLM. Second, also as indicated above, we have added experiments and analyses to Supp. Fig. 1 showing that noise correlations are similar between large-pixel checkerboard noise to a natural movie. Third, previous work in the primate retina shows that for RGCs, noise correlations are relatively robust to changes in the stimulus (see Fig. 5 of Greschner et al, 2011, J Neurophys). We now highlight these points in the Discussion (see Line 420)

Finally, we examined the effect of noise correlations on discriminating pixels in natural scenes using the model presented in Fig. 7. As a reminder, the analysis in Fig. 7 presents a checkerboard stimulus to a population of modeled RGCs whose responses on each simulated trial were given by: a) the overlap between their receptive fields and the bright pixels in the stimulus plus b) Gaussian (pseudo-Poisson) noise, which was potentially correlated between neurons. We then modified the noise correlations between these neurons, and studied how well the stimulus could be decoded from the neural responses. In our new analysis (Supp. Fig. 7 of the revision), we used binarized natural image stimuli. This analysis yielded very similar trends in population failure as when using checkerboard noise, indicating there is nothing specific or

particular about the population failure results for checkerboard noise stimuli, and that these results generalize to more natural stimulus conditions.

We hope the additions to Supp. Fig. 1, a new Supp. Fig. 7, and text changes to the manuscript address the concerns and questions expressed by the Reviewer.

Moderate:

Maybe this is more stylistic than substantive, but I think the comparison to single cells and the term “population failure” are a bit over-emphasized. It is indeed an interesting point that the failure of assuming independence can overwhelm the benefit of adding more cells to a decoder, but what is so special about this one vs. many comparison? It makes intuitive sense that adding what amounts to garbage information from surrounding cells because you have a very poor noise model could be worse than ignoring those cells completely. Calling population failure a “mode” seems to impart a false dichotomy on the situation. Instead, as the authors show, the reduction in information your decoder suffers by assuming the incorrect noise model varies smoothly as a function of the size and extent of the noise correlations. The fact that it becomes worse than a single cell decoder just means that it crossed a particular threshold – and this is very likely to depend on how much information single cells carry in the particular decoding task. Since the checkerboard squares were about the size of a whole RF center, it stands to reason that a single-cell decoder would perform at least decently well on the task. For a different task, like encoding the position of a spot, it may be impossible for a single-cell decoder to outperform a population code for the trivial reason of the limited size of a single RF.

We agree with the reviewer that the cost of ignoring noise correlations can be significant and can generalize to conditions where populations remain necessary for accurate decoding. We explore one such case in Supp. Fig. 4F-I where the task is to decode a spatial pattern of pixel intensities. Here a population of just 7 cells with knowledge of noise correlations outperforms a population of 18 cells that assumes independence (compare Supp. Fig. 4G and H). Ignoring correlations thus makes about half of the population (worse than) useless in this decoding task. Thus, the Reviewer is correct that population failure generalizes beyond the case of falling below the threshold performance of just one cell. We have added some text to the Discussion to clarify these points (lines 440-448).

At the same time, we respectfully disagree with the reviewer about the importance of seeing neuronal populations yield worse decoding than single neurons from those populations, when noise correlations are ignored. We have a few reasons for this assertion:

First, we think that a population of cells failing to reach the performance of a single cell is a reasonable and significant benchmark. Failing to reach this level of performance suggests that downstream brain areas ought not to combine information across cells, if the decoding procedure assumes those cells have independent noise. Furthermore, as noted above, it indicates that failing to account for noise correlations is often more costly than the total benefit of adding signals from nearby neurons and the associated signal correlations.

This assertion is supported by the responses of Reviewers 1 and 3. In fact Reviewer 3 wrote, “What is interesting is that the decrease of performance can be very large and is even larger than decoding from a single cell.” This result wasn’t intuitive for us (the authors) either, and when we have presented this result at conferences we have encountered a lot of interest and surprise at this observation. These reactions motivated the models in Figs. 6 and 7 to show how and when population failure can occur. Reviewer 3 requested a more thorough analysis of the model in Fig. 6, suggesting that the reviewer didn’t find the model or the results trivial.

Finally, it is important to note that the signals we are adding from the surrounding cells are not ‘garbage’. Nearly always, those cells -- as pointed out by the Reviewer -- have some of their receptive field overlapping with the decoded pixel, meaning their firing rate depends on the

intensity of that pixel. Thus, those surrounding cells carry a decodable signal about the activity of that pixel along with some amount of noise. The ‘independent’ model fails in decoding because it can only account for signal correlations, not noise correlations. The coupled decoder, on the other hand, is able to fully utilize the added information from surrounding cells, as demonstrated by its increased performance over the single cell model (Fig. 4; also see Supp. Fig. 4)

Minor:

A study from Michael Berry’s lab looked at decoding spatial structure from populations of RGCs and found a related result in that codes that assumed independence were incapable of reaching low error rates (Schwartz et al., 2012). The stimuli were different, and the focus on that work was on larger populations and on stimulus correlations. Still, it is worth discussing how the present results relate to that study.

Reference

Schwartz G, Macke J, Amodei D, Tang H, Berry MJ (2012) Low error discrimination using a correlated population code. *J Neurophysiol* 108:1069–1088 Available at:<http://www.pubmedcentral.nih.gov/articlerender.fcgi?artid=3424080&tool=pmcentrez&rendertype=abstract>.

We thank the reviewer for pointing us to this important work. We now include a brief discussion beginning on lines 366.

“For decoding, ignoring signal correlations in large populations of RGCs is particularly detrimental to decoding performance, especially when decoders have access to the fine temporal structure in spike trains {Schwartz2012}.”

Reviewer #3 (Remarks to the Author):

Ruda, Zylberberg and Field present an analysis of the impact of noise correlation in the activity of rat retinal ganglion cells across photopic and scotopic conditions. They combined multi-electrode array experiments with a computational analysis of the decoding performance based on GLM modeling. From abstract, result and discussion, the main results of the paper are two: (i) OFF-bt RGCs increase their noise-correlation in scotopic condition and this induces a large improvement in the decoding performance when noise-correlations are taken into account. (ii) assuming independent cells cause a drop in performance when decoding. There is a whole literature on how noise-correlations impact information transmission in neuronal systems, and in particular in the retina. This paper brings a contribution by focusing on the decoding, instead of the encoding. This point of view is not new, but of equal, if not larger, importance and surely less investigated than the other.

From a pure theoretical point of view, point (ii) is not particularly interesting. The fact that by decoding with a wrong model (independent GLM, figs 4&5 or independent linear model, figs 6&7) you get lower performance is not surprising. What is interesting is that the decrease of performance can be very large and is even larger than decoding from a single cell. The authors then construct two toy models and show that this effect depends on the ranges of parameters. For some ranges the difference is minimal, for others the improvement can be very large (population failure). So the question here is where the real retina lies in this scenario. This is

where it comes point (i): when applied to real data, and especially from scotopic regimes, the authors found a strong population failure (Fig 2), but only for OFF-bt cells and not for OFF-bs (Fig.3).

On a side note, and as the authors state in the discussion, the fact that noise correlations increase in scotopic regimes is not new and well documented in the literature. Similarly their results in the improvement of information transmission in photopic condition are consistent with previous findings.

I found both points (i) and (ii) interesting, but I don't think the obtained results are much novel and/or convincing to support publication, at least not in the present form, and I believe they should be developed further. In particular the role of the stimulus is quite absent in the author analysis, and this is a major lack when dealing with an early sensory system like the retina. More details in the following. Also, the fact that the impact on decoding performance by neglecting noise-correlations is very large in scotopic condition is interesting but it is also a direct consequence of the larger noise correlations in this regime.

Overall the paper is well written and easy to read. Good work here. Given the interdisciplinary topic here addressed, this was not easy.

We appreciate the Reviewer's comments and their thoughtful summary of the study. In what follows, we explain a number of stimulus and modeling manipulations we have added to the manuscript to show that these results generalize well to more natural stimulus conditions. We detail these changes in our point-by-point responses below.

The reviewer is correct that the larger consequence of ignoring correlations under scotopic conditions is more-or-less directly caused by the correlations being stronger under scotopic conditions. However, there are documented cases where weak correlations have a large impact on population codes (e.g. Hu, Zylberberg, Shea-Brown PLoS CB 2014 and Schneidman, Berry, Segev, Bialek, Nature 2006), so we think it didn't need to be the case that "strong correlations are important and weak ones are not important." Furthermore, the deep connection we show between light adaptation, correlation strength, and decoding performance is made nowhere in the literature (that we know of), and we think it is an important point that needs to be made somewhere prominently. After all, adapting to the 10-12 orders of magnitude in light intensities encountered in the natural environment is surely one of the most important visual encoding tasks accomplished by the retina. Furthermore, we think that most visual neuroscientists reasonably believe, based on previous studies, that noise correlations have a relatively minor (0-20%) impact on decoding performance in the retina. But we show this statement depends strongly on cell type and adaptation state. We are not aware of any previous studies that make this point. Finally, as the Reviewer points out, the observation of population failure is novel and surprising, which we think adds further justification for publishing this work in Nature Communications.

Major points

Toy model of Fig. 6, two neurons with gaussian noise responding to two stimuli. In this simple case most of the computation can be performed analytically using the fisher information and the d-prime approach (Seung and Sompolinsky 1993). Previous works in the literature have shown that the increase of information transmission or decoding depends on the stimulus, and in particular if stimulus and noise correlations have the same or opposite sign (differential-correlations in Moreno-Bote et al Nature 2014, see also Franke et al Neuron 2016). The fact

that the authors find a signature of population failure here is interesting, but I would require a full analysis of this simple model upon changing the strength of noise correlations, and the extent and sign of stimulus correlation. As the authors found, noise correlations are positive in the retina, but signal correlations can be both positive or negative depending on the stimulus. Strictly speaking, in your example both neurons have a larger firing rate for stimulus 2, what if only one does, but the other decreases its mean rate?

We thank the Reviewer for prompting us to explore and highlight this analysis and model further. We have added to Fig. 6 a more complete analysis of how decoding performance in this model depends on both signal and noise correlations. Specifically, we highlight the conditions under which we observe population failure for this 2-cell model with linear decoding when signal and noise correlations are positive and negative.

We also note that the most frequent case in which signals of nearby neurons are anti-correlated is between ON and OFF cells that have overlapping receptive fields. Previous work (e.g. Mastrorarde 1983, J Neurophys; DeVries 1999, J Neurophys; and Greschner et al 2011, J Neurophys) shows ON and OFF cells with overlapping receptive fields exhibit anti-correlated spiking, so the common 'direction' of the signal and noise correlations would remain the same in this case and our observation of 'population failure' would be preserved. This case is shown in Fig. 6G in the negative noise correlation area (left side) of the '- signal' bar: we observe that when ignoring strong, negative noise correlations, decoding from two cells with negative signal correlations exhibits population failure (represented by purple in the colormap).

From this simple drawing, it seems to me that the population failure should be much weaker, if not absent at all if the neurons are negatively stimulus-correlated. I think that the authors should take advantage of the possibility to perform all the computations analytically and present a complete picture on when and with what extent population failure happens.

We think that the Reviewer is also considering the noise correlations will remain positive when the stimulus correlations are negative. We agree with the reviewer that under that condition, population failure is not predicted by this simple model (see new Fig. 6G, top bar). However, as pointed out above, stimulus-induced correlations will very frequently be negative among ON and OFF RGCs with overlapping receptive fields: In this case, the noise correlations are also negative, and can be quite large because the two cells will share a lot of common input (indirectly) from photoreceptors. Negative stimulus and noise correlations produce a symmetric case with positive stimulus and noise correlations, and thus population failure would be expected to frequently occur under these conditions as well (shown in Fig. 6G, bottom bar). We now make this point: see lines 283-291.

The lack of the analysis of the role of stimulus is even more evident on the retinal results of Figs 4&5. The authors present retinal responses to checkerboard stimulation with checks of a rather large size of 240 microns. This stimulus induces strong stimulus correlation between nearby cells. Therefore it allows for analysing only a rather small spectrum of possible combinations of stimulus and noise correlations. In my opinion this limitation weakens strongly the validity of the authors conclusion, namely the fact that population failure is a relevant phenomenon for downstream neurons. In conclusion, I found this result not satisfying, and I would recommend enriching it by the analysis of other stimuli that can induce both positive and negative stimulus correlations, as for example drifting gabors. In the paper they are missing, but often multi-electrode array experiments also record ON cells. Do they show noise-correlations?

Also, do you find noise-correlations also between cells of different types?

We agree with the Reviewer that considering the generalizability of our results to other stimulus regimes, beyond changes in light level, is important. We have addressed this concern from both data and modeling perspectives.

First, we compare noise correlations measured during large-checker white noise to noise correlations during (1) spontaneous activity, (2) small checkerboard noise, and (3) a natural movie (Supp. Fig. 1). The natural movie consisted of a camera mounted to the head of a cat walking through a forest (Betsch et al 2004, Biol Cybern). We performed this comparison at both the rod and cone light levels. Across stimulus conditions, the noise correlations were similar. However, the noise correlations were somewhat higher under spontaneous activity, small checkerboard noise and the natural movie, than during the large checkerboard noise. Thus, our stimulus provides a conservative estimate on the strength of the noise correlations (Supp. Fig. 1B and D). Given our results in Figs. 6, 7, Supp. Fig. 5, and Supp. Fig. 7, these conservative estimates on the noise correlations will produce conservative estimates on the magnitude and frequency of population failure. We think the estimated noise correlations are smaller during the large-checkerboard noise because this stimulus drives strong responses in the neurons which pushes them toward saturation, thereby reducing the dynamic range upon which the noise correlations can be added to the signal. Similar effects have been observed previously (see Fig. 5 of Greschner et al 2011, J Neurophys).

To summarize, under more natural stimulus conditions (e.g. a natural movie) noise correlations are even a little stronger than our estimates using large checkerboard noise (see Supp. Fig. 1 B & E). Thus our estimates of the noise correlations likely represent a lower bound on their strength during natural vision. We discuss the implications of these observations in the Discussion, lines 420-488.

Second, the Reviewer is concerned that these checkers are particularly large. We point out that the area of each checker is actually similar to the area of the receptive field center for the RGCs that we are using to decode (See Figs. 3A, 4A, and 5A for examples of RFs and the pixel sizes that were decoded). The RFs had an average diameter of 259 μm (at 1-sigma) and the pixels had an area of 252 μm^2 on each edge. Second, the size of these checkers is only ~4-fold higher than the spatial resolving acuity of the rat visual system, which is 1 cycle/deg (Prusky et al., 2000, Vision Research), which corresponds to ~60 $\mu\text{m}/\text{deg}$ on the retina. The linear size of the checkers is 252 μm , which is 60 x 4.2.. Thus, we think the parameters of this stimulus we used are a reasonable match to the RFs sizes we are examining and the acuity of the rat's visual system.

Third, we have examined how decoding natural scenes influences our predictions of population failure using a model of the RGC array: see the new Supp. Fig. 7. In this model we can explicitly control the amount of signal correlation via the amount of receptive field overlap, the response polarity, the strength of noise correlations and the spatial extent of noise correlations. Previously, this model only analyzed the discrimination performance for checkerboard patterns (Fig. 7). We have added an analysis of model performance for natural scenes. This analysis shows that the conditions and severity of population failure predicted by the model are minimally altered by decoding natural scenes versus checkerboard patterns (compare Fig. 7 to Supp. Fig. 7).

The reviewer further raises several questions about cells of different types, especially ON cells. ON types do show similar noise correlation structures to the OFF data we have presented. We have added a paragraph to the first subsection of the Discussion to explain our choice for focusing on OFF RGCs (their mosaics were the most complete and their spikes were well isolated). That paragraph begins on line 280. We have also added a phrase on line 373 to make it clear that the cell types and species used across our study, the Pillow et al 2008, Nature and the Meytlis et al 2012, Vision Res studies are distinct.

Because we focused on cells of the same type, which have very similar receptive fields aside from displacements in space, stimulus correlations are generally positive. As noted above, negative stimulus correlations occur most frequently for cell pairs with different polarities of visual sensitivity (e.g. ON versus OFF RGCs). We agree with the reviewer that negative stimulus correlations is an interesting space to explore, but it is mostly relevant for correlations among cells of different types. We think across-type correlations is beyond the scope of the current manuscript because there are some complicating factors with decoding from multiple cell types:

1. It is very difficult to control for the number and alignment of cells of each type across different local groups of RGC; the possible cell type-pair combinations from just the 13 types we can identify create a huge analysis space;
2. Better spike sorting is required because cells of different types are not regularly spaced, so their spikes are much more frequently overlapping in time on the same electrodes (known as the 'superposition' problem in spike sorting);

We plan to follow up on this study by considering across-type correlations in the future with more advanced spike sorting techniques and we hope to fit those results within a broader framework of the relationship between signal and noise correlations in visual coding. We hope these responses clarify the manuscript and address the Reviewer's comments.

Minor points

Figure 1. Even if partially shown in figure 2, it would be interesting here to see the 6 PSTHs of cells i,ii,iii in the two conditions.

What is the time-bin length for computing the cross-correlation of panel D? If it is of the order of 10ms, it would be nice to see, maybe in an inset, the peak of the correlations at finer time-scale (1-2ms).

How much of the changes in the cross-correlation between photopic and scotopic conditions are due to changes in the auto-correlation of individual neurons?

We have added the PSTHs to Fig. 1. The time bin for computing cross-correlograms is 5ms, and we did not see large differences between 1-5ms binning. We have updated the cross-correlograms to be normalized by the autocorrelation of each cell (see Methods Line 574). The autocorrelation functions are relatively stable with the exception of a change in the mean spike rate. We quantified the change in autocorrelation shape by taking the dot product between a cell's autocorrelation at each light level. For all 102 OFF-bt RGCs used in this study, the average dot product is 0.91 ± 0.005 s.e.m.

Figure 2. Panel D, why did you show the data from only one retina?

Panel F. it seems that you over-estimate the CCF peak in photopic condition, but it is hard to see it clearly. Could you split this panel in two, one panel for each condition please?

How much differ the GLM filters in the two conditions? Can you compare them please?

We have changed Fig. 2D to show results from all retinas. We have also split Fig. 2F into two panels for visual clarity. To compare GLM filters across the light levels, we show all example filters (receptive field, spike-history, and coupling) in Supp. Fig. 2. CCFs peaks are sometimes over-estimated a bit by the coupled GLM fits in the photopic condition.

Decoding. I found that some important details are missing on how the decoding model is constructed. The author explains that they use optimal bayesian decoding to predict stimulus

value in one pixel for 6 temporal windows. However they do not explain clearly what are the input of the decoder. From figure 3, I understood that it is fed with the spiking times of the small group of cells, but what about the other stimulus pixels around the central one? Do you need them?

Also, it is not clear to me if you decoded the stimulus only in cases where a central cell has a large overlap with the pixel. If it is the case, how many pixels over the total were you able to decode?

We have edited the methods section on decoding to make these points clearer--see lines 607-611. To answer the Reviewer's specific questions, the decoder is provided with the intensity values of the non-decoded pixels. This matches the procedures of previous work (e.g. Pillow et al 2008, Nature; Meytlis et al 2012, Vision Res) to make our results maximally comparable. Decoding could be performed without providing the decoder the values of the other pixels, but overall performance will decrease.

For single pixel decoding over 6 movie frames (the main condition we use to decode in the manuscript), we chose the pixel with the highest weight in the STA for a given RGC. Our analysis covers 72 unique pixels across 4 experiments (retinas).

Figure 4. I have hard times in understanding what is plotted in panel D

We have added panel insets to help clarify what is being plotted here and additional explanation in the legend of Fig. 4.

Figure 6. Following the first major point, I suggest adding an example of negatively stimulus-correlated neurons (example the toy picture above). Limits of x-axis in panel A&B and D&E should be the same for readability

We have made the Reviewer's suggested changes and additions to this figure.

Figure 7. Could you add panels also for the single cell decoder please?

We could, but the single cell model doesn't change with changes in the correlation structure: its performance is indicated by the white color in the green/magenta colormaps.

Supp fig 2:

*Panel C&D. Left, why does the independent GLM have higher performance than the coupled one?

We think this is because the coupled model is a bit more prone to overfitting (more parameters to fit).

*Right. Is this the same metric from Fig.2D? If not, it should be better clarified.

Yes, the metric is the same. We clarify this in the caption to Supp. Fig. 2.

*Panel F, as in Fig2, this is hard to see, you should split the two conditions.

We have done as requested by the reviewer.

L191-192 To which paragraph the "see methods" refers to?

We apologize that a clear statement about this did not make it into the Methods. We have added the text at line 597 to clarify this procedure.

REVIEWERS' COMMENTS:

Reviewer #1 (Remarks to the Author):

The authors have addressed all of my previous feedback to satisfaction. I particularly like the addition of the use of binarized natural scenes in their model to support their analyses and interpretation of the results on the checkerboard stimuli.

Overall, I think the revised manuscript is a strong contribution to our understanding of sensory coding across different environmental conditions and how the nervous system must deal with noise correlations changing as these environmental conditions change.

I therefore firmly support the publication of this manuscript in Nature Communications.

Best,

--Braden Brinkman

Reviewer #2 (Remarks to the Author):

The authors have generated a comprehensive and satisfactory response to my critiques from the first round of review, including new experiments with binarized natural movies, new analyses, and improvements in the presentation of some of the data and the clarity of the writing in several places. In particular, it was important to show that the results obtained with 250 micron checkers generalize to more natural stimuli, which the authors have now done. My misunderstanding about details of the work regarding stimulus vs. noise correlations has been clarified. This is a nice contribution to the field of the role of correlations in the retinal code. Hopefully it will be extended to other RGC types in the future.

Reviewer #3 (Remarks to the Author):

In the revised version of their manuscript, Ruda, Zylberberg and Field have fully addressed all my major comments, and in my opinion also those of the other referees. Their arguments are solid, the paper has been substantially improved thanks to novel data and analyses, and I'm now convinced that the paper deserves publication in Nature Communications.

In line with the other referees, I asked the authors to test if the population failure effect arises also in the case of more rich and natural stimuli. To prove this the authors have first verified that in response to natural stimuli noise-correlations are of similar if not greater strength. For this they have added novel data from a third retina stimulated with a natural video. Then, instead of applying a GLM model analysis on these responses, which would have been very hard if not impossible (Heitman et al., 2016), they have used their retinal linear toy-model. This analysis shows that population failure happens also for natural stimuli. Even if these results come from a toy-model, I believe that they can be trusted with more than acceptable confidence. In addition, I don't think it would be possible to improve this without fully developing a more powerful model, which is beyond the scope of the paper.

The authors have also addressed my concerns about the toy picture of Fig6, and showed clearly what should be expected in the case of 'opposite sign' between signal and noise correlations.

It is unfortunate that the authors refuse to include an analysis of the effects of noise correlation between rgc-pair of different types. I understand that experimentally the necessary data are much harder to collect. I also understand that adding these analyses might have made the paper too rich

and dense. For these reasons, I don't think the lack of these analyses should prevent publication.

I have only some minor comments:

- L 68: we recorded RAT RGC responses...
- Fig 1: cell ii → cell iii
- L328-329 can you add a quantification of the difference in the power spectrum?
- I believe that the results for natural stimuli deserve more space in the main paper. They are very interesting but somehow hidden in the supplementary figures
- Should we expect population failure also beyond the retina? The authors might add a paragraph in the discussion.
- Can the authors add comments on what would be the results if the decoded pixel lies in between two RGC centers?

Nice work,
Congrats.

Response to reviewers

REVIEWERS' COMMENTS:

Reviewer #1 (Remarks to the Author):

The authors have addressed all of my previous feedback to satisfaction. I particularly like the addition of the use of binarized natural scenes in their model to support their analyses and interpretation of the results on the checkerboard stimuli.

Overall, I think the revised manuscript is a strong contribution to our understanding of sensory coding across different environmental conditions and how the nervous system must deal with noise correlations changing as these environmental conditions change.

I therefore firmly support the publication of this manuscript in Nature Communications.

Best,

--Braden Brinkman

We thank the reviewer for their time and comments, which improved the study and the manuscript.

Reviewer #2 (Remarks to the Author):

The authors have generated a comprehensive and satisfactory response to my critiques from the first round of review, including new experiments with binarized natural movies, new analyses, and improvements in the presentation of some of the data and the clarity of the writing in several places. In particular, it was important to show that the results obtained with 250 micron checkers generalize to more natural stimuli, which the authors have now done. My misunderstanding about details of the work regarding stimulus vs. noise correlations has been clarified. This is a nice contribution to the field of the role of correlations in the retinal code. Hopefully it will be extended to other RGC types in the future.

We thank the reviewer for their time and comments, which improved the study and the manuscript.

Reviewer #3 (Remarks to the Author):

In the revised version of their manuscript, Ruda, Zylberberg and Field have fully addressed all my major comments, and in my opinion also those of the other referees. Their arguments are solid, the paper has been substantially improved thanks to novel data and analyses, and I'm now convinced that the paper deserves publication in Nature Communications.

In line with the other referees, I asked the authors to test if the population failure effect arises also in the case of more rich and natural stimuli. To prove this the authors have first verified that in response to natural stimuli noise-correlations are of similar if not greater strength. For this they have added novel data from a third retina stimulated with a natural video. Then, instead of applying a GLM model analysis on these responses, which would have been very hard if not impossible (Heitman et al., 2016), they have used their retinal linear toy-

model. This analysis shows that population failure happens also for natural stimuli. Even if these results come from a toy-model, I believe that they can be trusted with more than acceptable confidence. In addition, I don't think it would be possible to improve this without fully developing a more powerful model, which is beyond the scope of the paper.

The authors have also addressed my concerns about the toy picture of Fig6, and showed clearly what should be expected in the case of 'opposite sign' between signal and noise correlations.

It is unfortunate that the authors refuse to include an analysis of the effects of noise correlation between rgc-pair of different types. I understand that experimentally the necessary data are much harder to collect. I also understand that adding these analyses might have made the paper too rich and dense. For these reasons, I don't think the lack of these analyses should prevent publication.

I have only some minor comments:

- L 68: we recorded RAT RGC responses...

We specify the species on the same line: "We recorded RGC responses across a range of light intensities from segments of rat retina on a large-scale MEA."

- Fig 1: cell ii → cell iii

Thanks for catching this. We've fixed the label.

- L328-329 can you add a quantification of the difference in the power spectrum?

We added this power spectrum to Figure 7, panel H.

- I believe that the results for natural stimuli deserve more space in the main paper. They are very interesting but somehow hidden in the supplementary figures

We thank the reviewer for this comment. We have moved the results pertaining to natural stimuli from Supp. Fig. 7 to the main Fig. 7.

- Should we expect population failure also beyond the retina? The authors might add a paragraph in the discussion.

We have added a paragraph to the end of the Discussion on how population failure might generalize beyond the retina: see Lines 478-488.

- Can the authors add comments on what would be the results if the decoded pixel lies in between two RGC centers?

It is difficult to add a simple statement addressing this question. If a decoded pixel lies between two receptive fields, there are likely to be stronger signal correlations between the two RGCs. However, the effect on decoding while ignoring noise correlations ultimately depends on the strength of the noise correlations relative to this signal correlation. In our analyses, we have averaged over a range of pixel-receptive field alignments that occurred in our recordings.

Nice work,
Congrats.

We thank the reviewer for their time and comments, which improved the study and the manuscript.